# Dopamine-dependent prefrontal reactivations explain long-term benefit of fear extinction

A.M.V. Gerlicher[1,2,4], O. Tüscher[2,3] & R. Kalisch[1,2]

Fear extinction does not prevent post-traumatic stress or have long-term therapeutic benefits in fear-related disorders unless extinction memories are easily retrieved at later encounters with the once-threatening stimulus. Previous research in rodents has pointed towards a role for spontaneous prefrontal activity occurring after extinction learning in stabilizing and consolidating extinction memories. In other memory domains spontaneous post-learning activity has been linked to dopamine. Here, we show that a neural activation pattern — evoked in the ventromedial prefrontal cortex (vmPFC) by the unexpected omission of the feared outcome during extinction learning — spontaneously reappears during postextinction rest. The number of spontaneous vmPFC pattern reactivations predicts extinction memory retrieval and vmPFC activation at test 24 h later. Critically, pharmacologically enhancing dopaminergic activity during extinction consolidation amplifies spontaneous vmPFC reactivations and correspondingly improves extinction memory retrieval at test. Hence, a spontaneous dopamine-dependent memory consolidation-based mechanism may underlie the long-term behavioral effects of fear extinction.

[1] Neuroimaging Center (NIC), Focus Program Translational Neuroscience (FTN), Johannes Gutenberg University Medical Center, Langenbeckstr. 1, 55131 Mainz, Germany. [2] Deutsches Resilienz Zentrum (DRZ), Johannes Gutenberg University Medical Center, Untere Zahlbacher Str. 8, 55131 Mainz, Germany. [3] Department of Psychiatry and Psychotherapy, Johannes Gutenberg University Medical Center, Untere Zahlbacher Str. 8, 55131 Mainz, Germany. [4] Present address: Department of Clinical Psychology, University of Amsterdam, Nieuwe Achtergracht 129B, 1018 WS Amsterdam, The Netherlands. Correspondence and requests for materials should be addressed to A.M.V.G. (email: a.m.v.gerlicher@uva.nl)

Fear extinction is believed to protect against the development of stress-related pathology after trauma[1]. Furthermore, exposure interventions based on the principles of extinction learning are a cornerstone of cognitive-behavioral therapy of anxiety disorders and post-traumatic stress disorder[2]. During fear extinction, conditioned fear responses (CRs) are reduced through repeated exposure to the fear-inducing stimulus (conditioned stimulus, CS) in the absence of the aversive outcome (unconditioned stimulus, US) with which it had previously been paired. Extinction learning establishes a "CS–noUS" association or extinction memory that needs to be retrieved at later confrontations with the CS in order to inhibit the expression of the original "CS–US" association or fear memory and prevent the return of CRs[3]. An important problem that plagues extinction research is that the success of extinction learning is neither a strong nor reliable predictor of the long-term retrieval of the extinction memory[4,5]. Even complete abolishment of CRs in an extinction session does not guarantee the absence of CRs at later test. Conversely, no or incomplete extinction may still protect against the return of CRs[4,5]. We here suggest that spontaneous, stimulus-independent neural consolidation processes occurring in the hours or days after learning[6] can help to explain whether extinction memories are successfully retrieved and prevent the return of fear in the long-term.

There is increasing evidence that neural consolidation processes may not simply be a deterministic prolongation of encoding processes but may, instead, make an independent contribution to memory formation[7]. In rodents, extinction memory consolidation depends on the infralimbic part of the medial prefrontal cortex (IL)[8–10]. The IL exhibits neural firing activity in the form of spontaneous gamma-frequency bursts in the hours after extinction[8]. The number of IL bursts predicts extinction memory retrieval at test 24 h later, such that animals with more bursts during consolidation show smaller CRs at test[8]. IL bursts have also been observed in response to CS presentations during extinction learning[11], raising the possibility that stimulus-evoked neural activity patterns established during extinction learning may be reactivated in a stimulus-independent fashion after learning. The first aim of the study was, thus, to examine whether spontaneous reactivations of extinction-related activity patterns in the ventromedial prefrontal cortex (vmPFC) (the human homolog of the IL[12]) contribute to human extinction consolidation and can predict the long-term expression of fear.

Furthermore, extinction memory consolidation depends on dopaminergic activity in the prefrontal cortex. Blocking dopamine receptors before or after extinction in the IL impairs extinction consolidation and leads to a return of CRs[13,14], while increasing dopaminergic receptor activity during or after extinction leads to improved extinction memory retrieval[15–18]. Two studies suggest that extinction memory consolidation can also be enhanced in humans by systemic postextinction administration of the dopamine precursor levodopa[17,19]. Interestingly, dopamine has been shown to amplify spontaneous burst firing in the prefrontal cortex in rodents[20]. In addition, dopamine was shown to increase the reactivation of learning-related neural activity during consolidation in the domain of spatial memory[21]. Based on these findings, the second aim of the study was to test whether a postextinction enhancement of dopaminergic activity increases vmPFC reactivations of extinction-related activity patterns after learning and thereby improves extinction memory retrieval.

We tested these hypotheses in $n = 40$ male human participants (Supplementary Table 1) in a functional magnetic resonance imaging (fMRI) experiment, with fear conditioning on day 1, extinction training immediately followed by the oral administration of either placebo or 150/37, 5 mg levodopa/benserazide (L-DOPA) on day 2, and retrieval test on day 3 (Fig. 1). During

conditioning, one of two geometric symbols (CS+) was paired with a painful stimulation (US) in 50% of trials, while the other symbol (CS−) was never paired with the US. During extinction and test, both symbols were presented in the absence of the US. During all experimental phases, we recorded skin conductance responses (SCRs) to both CSs as index of the CR. In addition, US-expectancy ratings to both CSs were assessed before and after each experimental phase to assess contingency awareness (Supplementary Fig. 1). In order to capture spontaneous consolidation-related neural activity, resting-state fMRI scans were collected 10, 45, and 90 min after the end of the extinction session (Fig. 1a).

Our results show that the number of postextinction reactivations of an extinction learning-related vmPFC pattern (day 2) predicts extinction memory retrieval at test (day 3). As expected, postextinction administration of L-DOPA enhances extinction memory retrieval and reduces the expression of fear at test on day 3. Critically, L-DOPA also increases the number of vmPFC pattern reactivations, and this effect mediates the effect of L-DOPA on extinction memory retrieval. These results provide the first evidence for a critical role of dopamine-dependent vmPFC activity pattern reactivations in the consolidation of human extinction memories. Lastly, we show that spontaneous vmPFC pattern reactivations can explain long-term extinction memory retrieval above and beyond a marker of extinction learning success.

## Results

**Postextinction L-DOPA administration enhances extinction memory retrieval.** On day 1, CRs were successfully acquired, as indicated by greater SCRs to the CS+ than to the CS− in both groups (repeated-measures ANOVA: stimulus: $F_{1,35} = 57.95$, $P < .001$, partial $\eta^2 = 0.62$, stimulus × group: $P = 0.67$; $n = 37$ participants with sufficient SCR data quality, see Methods). On day 2, CRs were successfully retrieved at the beginning of extinction in both groups (stimulus: $F_{1,34} = 29.87$, $P < .001$, partial $\eta^2 = 0.47$; stimulus × group: $P = 0.36$; $n = 36$). At the end of extinction learning, there was still a significant CS+ > CS− difference on SCRs that did, however, not differ between groups (stimulus: $F_{1,34} = 14.49$, $P < 0.001$, partial $\eta^2 = 0.30$; stimulus × group: $P = 0.54$; $n = 36$). As predicted, the postextinction administration of L-DOPA compared to placebo (see Fig. 1a) enhanced extinction memory retrieval and reduced the expression of fear at test on day 3 (stimulus: $F_{1,33} = 44.61$, $P < .001$, partial $\eta^2 = 0.58$, stimulus × group: $F_{1,33} = 6.58$, $P = 0.02$, partial $\eta^2 = 0.17$; $n = 35$; Fig. 1b). Specifically, L-DOPA-treated participants showed significantly smaller SCRs to the formerly reinforced CS+ than placebo-treated participants (two-sample $t$ test: $T_{33} = -2.25$, $P = 0.03$, Cohen's $d = 0.74$; $n = 35$; Fig. 1b). Thus, in line with previous findings[17,18], a postextinction administration of L-DOPA compared to placebo (see Fig. 1a) enhanced extinction memory retrieval and reduced the long-term expression of conditioned fear.

**Reactivations of a CS+ offset-related vmPFC pattern predict extinction memory retrieval.** Next, we tested the first hypothesis of whether spontaneous reactivations of extinction learning-related neural activity in the time after extinction learning predict extinction memory retrieval. Due to its indispensable role in extinction memory consolidation[8–10], we focused our analyses on the vmPFC and specifically on the possible reactivation of an extinction learning-related vmPFC activity pattern during the postlearning resting-state scans (R in Fig. 1a). We first determined the spatial distribution of fMRI activity within the vmPFC

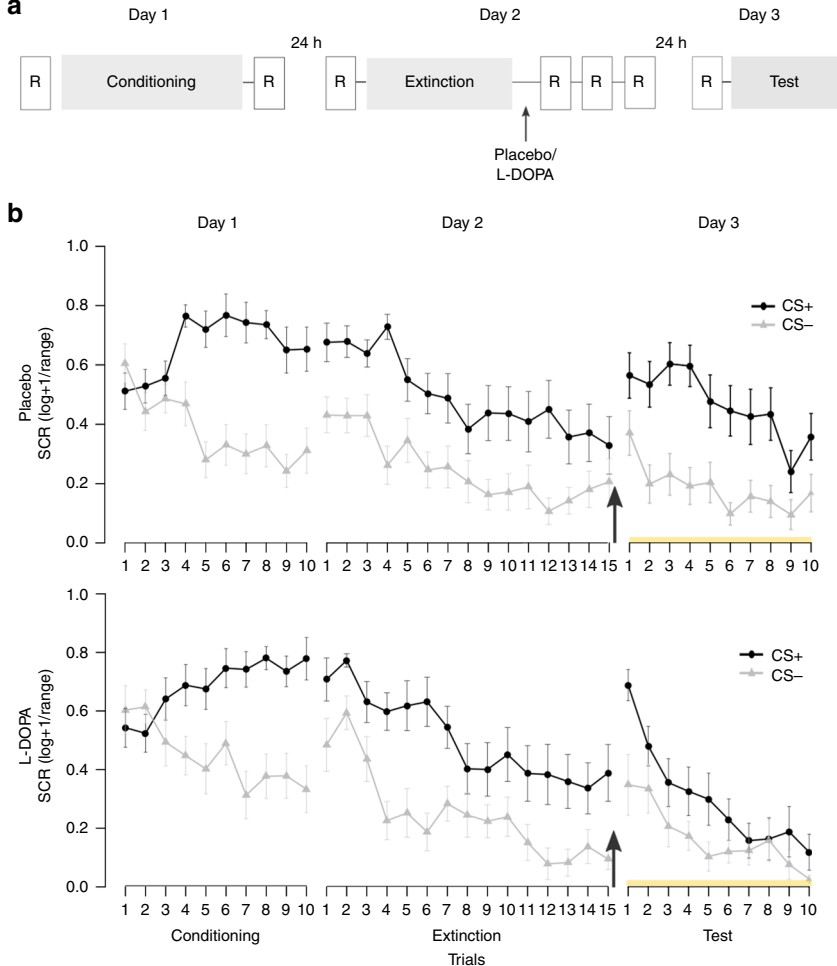

**Fig. 1** Experimental design and skin conductance responses. **a** Participants underwent a 3-day fMRI study with fear conditioning on day 1, extinction and subsequent placebo or L-DOPA administration on day 2 and test on day 3. During fear conditioning one of two geometric symbols (CS+) was reinforced with a painful electrical stimulation, while the other symbol (CS−) was never reinforced. Postextinction placebo or L-DOPA administration was randomized and double-blinded (placebo: $n = 20$, L-DOPA $n = 20$, all male, for group characteristics, see Supplementary Table 1). Resting-state fMRI scans (R) were acquired before and after fear conditioning, before and 10, 45, and 90 min after extinction, and before test. **b** During all experimental phases, we assessed conditioned responses (CRs) as skin conductance responses (SCRs) to CS+ and CS−. Upper panel depicts mean SCR to CS+ and CS− for placebo-, lower panel for L-DOPA-treated participants. The groups differed significantly on mean SCRs across the test phase on day 3 (marked by yellow line) due to significantly smaller mean SCRs to the CS+ in L-DOPA compared to placebo-treated participants. Note, that the group difference stemmed from significantly smaller CS+ evoked SCRs averaged across the whole test phase, but the speed of re-extinction did not differ significantly between drug groups (control analysis with stimulus (CS+, CS−) and trial (1–10) as within-, and group (placebo, L-DOPA) as between-subject factor: stimulus × group, $F_{1,33} = 6.58$, $P = 0.02$, partial $\eta^2 = 0.17$; stimulus × trial × group, $F_{9,297} = 1.32$, $P = 0.23$; $n = 35$). Data are presented as mean ± standard error of the mean (s.e.m)

at the critical time point for learning in the extinction session: the omission of the US at CS+ offset in early extinction trials.

Early in the extinction session, US omission is unexpected to the participants and elicits a prediction error. Prediction errors drive the correction of the US prediction associated with the CS and induce extinction learning[22]. To capture this time point, we extracted the average multivoxel vmPFC activity pattern evoked by the first five offsets of the CS+ (the CS that had previously been paired with the US during conditioning) during extinction. We then correlated each participant's CS+ offset-related vmPFC pattern with his individual vmPFC pattern from each imaging time point (volume) during the pre- and postextinction resting-state scans (Fig. 1a). As in ref. [23], the individual Pearson correlation values for each volume were z-transformed, using the mean and standard deviation of correlations from the pre-extinction scan as a baseline reference, to then threshold them at a cut-off of $Z = 2$ (~$P = 0.05$)[23]. On this basis, the suprathreshold correlations in the postextinction scans depicted in Fig. 2a can be

considered to reflect potential spontaneous reactivations of the CS+ offset-related vmPFC activity pattern. Pattern reactivations in the vmPFC were approximately uniformly distributed over the time courses of each postextinction resting-state scan (Fig. 2a), and only a minority of participants did not exhibit any potential reactivation in a given resting-state scan (10 min: $n = 1$, 45 min: $n = 3$, 90 min: $n = 1$).

Critically, the number of potential vmPFC pattern reactivations during the 45-min postextinction scan negatively predicted CRs at test on day 3 (multiple linear regression: $\beta_{45\ min} = -0.13$, SE = 0.03, $T_{30} = -4.05$, $P = 0.0003$; $n = 35$; Fig. 2b; for control analyses, see Supplementary Figs. 2, 3). That is, more CS+ offset-related vmPFC reactivations were associated with improved extinction memory retrieval (i.e., reduced CRs) at test. Reactivations during the scans 10 or 90 min after extinction did not significantly predict CRs ($\beta_{10\ min} = -0.01$, SE = 0.04, $T_{30} = -0.17$, $P = 0.86$; $\beta_{90\ min} = 0.06$, SE = 0.04, $T_{30} = 1.71$, $P = 0.10$).

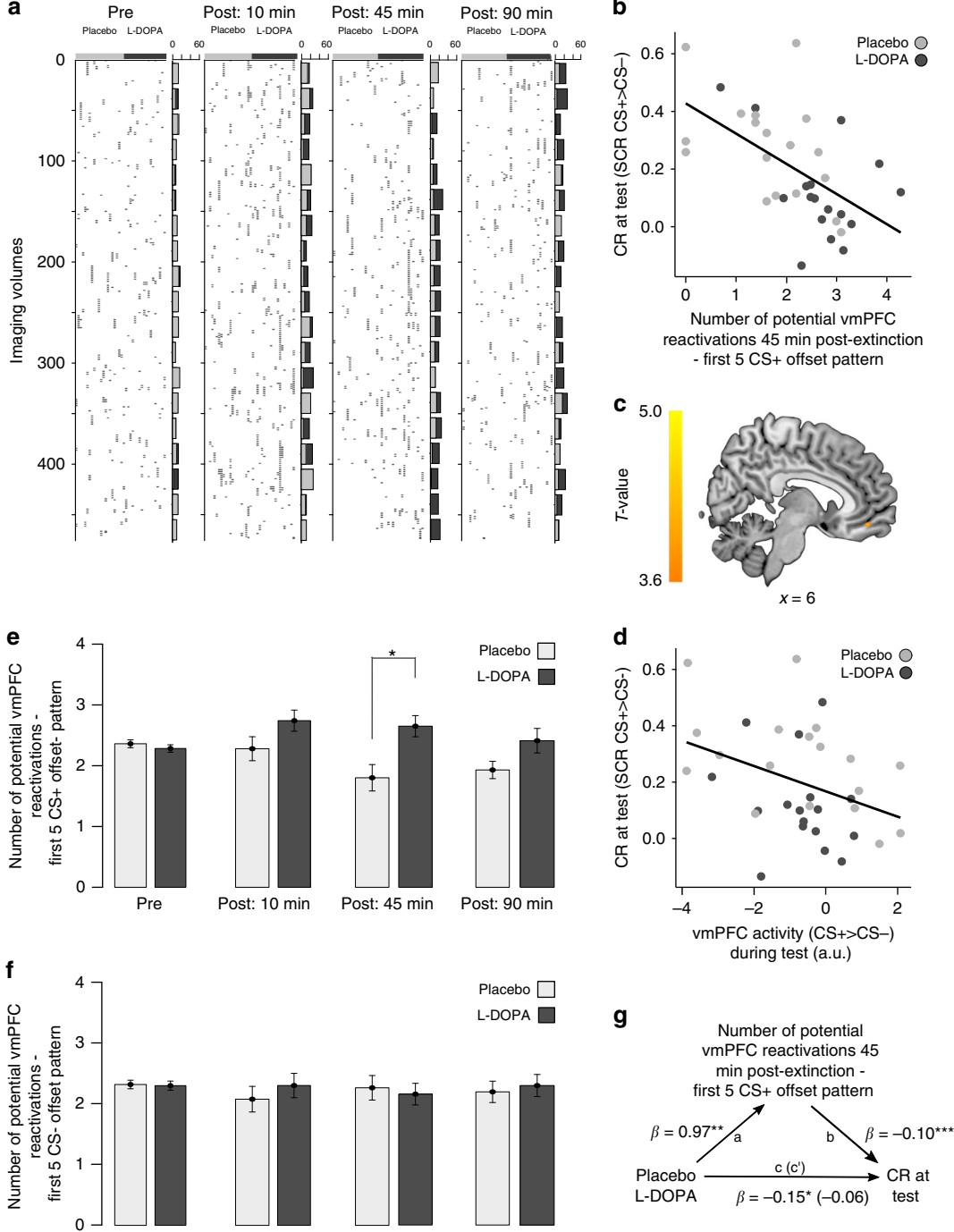

The administration of L-DOPA did not affect the predictive relationship between number of CS+ offset-related vmPFC reactivations and CRs at test, but comparable relationships between number of vmPFC reactivations and CRs at test were found in each group (multiple linear regression: interaction reactivations × group, $P = 0.99$; simple slope analysis placebo: $\beta = -0.09$, $P = 0.04$, L-DOPA: $\beta = -0.09$, $P = 0.07$; $n = 35$; Fig. 2b).

In addition, number of reactivations 45 min after extinction also predicted greater recruitment of the vmPFC during test on day 3, as determined in a canonical mass-univariate fMRI analysis (Fig. 2c). Activity in the vmPFC at test is an established neural correlate of extinction memory retrieval[24,25]. In line with this interpretation, vmPFC activity at test showed a negative

relationship to CRs at test in the present data, with greater vmPFC activity being associated with smaller differential CRs at test (Fig. 2d).

The finding that postextinction reactivations of the unexpected US omission pattern predict CRs at test was specific to resting-state activity in the vmPFC as opposed to all control regions (Supplementary Fig. 4). Importantly, the relationship was also specific to the pattern elicited by the unexpected US omission at early CS+ offset, as opposed to the expected US omission at early CS− offset (Supplementary Fig. 5a), the US omissions at CS+ offset during later phases of extinction (middle and last five trials; Supplementary Fig. 5b, c) at which the omission of the US was not surprising to the participant anymore, or the CS+ *onset* early, middle or late during extinction (Supplementary Fig. 5d–f). Even

**Fig. 2** Spontaneous postextinction reactivations support extinction memory consolidation. **a** Temporal distribution of suprathreshold correlations between the spatial fMRI activity pattern in vmPFC evoked on day 2 during extinction by the unexpected US omission (CS+ offsets in early extinction, i.e., first five trials) and the vmPFC pattern occurring at each resting-state volume after extinction on the same day. Each column represents one participant. The bars to the right of each panel represent sum scores per 25 imaging volumes, for both the placebo (light gray) and the L-DOPA group (dark gray). **b** Relation between the number of potential spontaneous CS+ offset-related vmPFC pattern reactivations 45 min after extinction on day 2 (here and in further graphs expressed as log + 1; for control analyses on nonlog-transformed data, see Supplementary Figs. 2, 3) and CRs at test on day 3 in the whole sample. **c** The number of potential vmPFC pattern reactivations 45 min after extinction on day 2 predicts CS+ > CS− evoked activity in vmPFC during test on day 3 (SPM multiple regression: MNI $x,y,z = 6,46,-14$; $Z = 3.86$, $P = 0.01$; small-volume (SVC) and family-wise error (FWE) corrected; $n = 40$). Display threshold $P < 0.05$, SVC, FWE, no masking applied. **d** Relation between vmPFC activity and CRs at test on day 3 (Pearson correlation: $r_{35} = -0.37$, $P = 0.03$; $n = 35$). **e** Effect of L-DOPA administration after extinction on day 2 on the number of spontaneous reactivations of CS+ offset-related vmPFC patterns during subsequent resting-state scans. **f** There was no effect of L-DOPA on number of spontaneous reactivations of CS− offset-related vmPFC patterns during resting-state scans on day 2 (repeated-measures ANOVA: time × group: $P = 0.63$, group: $P = 0.77$; $n = 40$). **g** The postextinction administration of L-DOPA had a significant positive effect on number of potential CS+ offset-related vmPFC reactivations (path a: $\beta = 0.97$, SE = 0.30, $T_{33} = 3.23$, $P = 0.003$; $n = 35$). The number of potential CS+ offset-related vmPFC reactivations 45 min after extinction was significantly negatively related to smaller differential CRs at test on day 3 (path b: $\beta = -0.10$, SE = 0.03, $T_{33} = -4.26$, $P = 0.0002$; $n = 35$). After inclusion of number of potential CS+ offset-related vmPFC reactivations into the latter model, the significant effect of drug on CRs at test (path c: $\beta = -0.15$, SE = 0.06, $T_{33} = -2.57$, $P = 0.02$; $n = 35$) decreased (path c′: $\beta = -0.06$, SE = 0.06, $T_{32} = -0.98$, $P = 0.33$; $n = 35$), indicating that the effect of L-DOPA on CRs at test on day 3 was significantly mediated (c–c′ = −0.09, 95% CI: −0.13 to −0.02, $P = 0.007$; bootstrapping procedure with 10,000 simulations; $n = 35$) by number of potential CS+ offset-related vmPFC reactivations 45 min after extinction

though the relationship was specific to the pattern elicited by the unexpected US omission at CS+ offset early during extinction, the exact number of trials used for estimating the early CS+ offset-related vmPFC activity pattern (here: first five CS+ offsets) was not critical for the results (Supplementary Figure 6). Lastly, the results are not dependent on cleaning or not-cleaning the resting-state BOLD activity time courses from nuisance signals (i.e., cerebrospinal fluid, white matter, and head motion) before identifying potential vmPFC reactivations (Supplementary Figure 7). Together, these findings suggest that spontaneous reactivations of a CS+ offset-related vmPFC activity pattern in the vmPFC after learning contribute to the consolidation of extinction memories and promote their later retrieval.

**The effect of L-DOPA on extinction memory retrieval can be explained by its effect on vmPFC reactivations.** Next, we turned to the second hypothesis concerning the potential role of dopamine in spontaneous pattern reactivations during consolidation and asked whether L-DOPA exerts its consolidation-enhancing effect by amplifying the frequency of postextinction vmPFC reactivations. Indeed, postextinction L-DOPA administration significantly increased the number of potential CS+ offset-related vmPFC pattern reactivations (time × group: $F_{3,114} = 3.51$, $P = 0.02$, partial $\eta^2 = 0.09$; $n = 40$), particularly 45 min after extinction and there was still a near-significant trend 90 min after extinction (post hoc two-sample t test; pre: $P = 0.39$; post: 10 min: $P = 0.09$; 45 min: $T_{38} = -3.05$, $P = 0.004$, Cohen's $d = 0.97$; 90 min: $T_{38} = -1.95$, $P = 0.06$, Cohen's $d = 0.62$; $n = 40$; Fig. 2e). This timing is consistent with the reported peak plasma levels of dopamine after oral L-DOPA intake[26].

This result cannot be explained by a potential diffuse effect of L-DOPA on the detection rate of any kind of potential reactivations, as L-DOPA did not affect reactivations of any control pattern such as the CS− offset-related patterns (Fig. 2f) or CS+ offset-related patterns elicited later during extinction learning (time × group: all $Ps > 0.25$). Intriguingly, the number of reactivations of the CS+ offset-related vmPFC pattern at 45 min postextinction mediated the effect of L-DOPA on CRs at test (Fig. 2g; mediation analysis—indirect mediator effect: $\beta = -0.09$, 95% CI: −0.13 to −0.02, $P = 0.007$; bootstrapping procedure with 10,000 simulations; $n = 35$). That is, L-DOPA affected fear at test by increasing the number of vmPFC reactivations during extinction memory consolidation.

**Spontaneous postextinction vmPFC reactivations explain extinction memory retrieval independent of extinction learning success.** Lastly, we asked whether potential CS+ offset-related vmPFC pattern reactivations explain extinction memory retrieval over and above interindividual differences in extinction learning itself, using hierarchical multiple linear regression. The best behavioral predictor of extinction retrieval we could identify in our extinction learning data (Supplementary Fig. 8) was the differential CR at the end of extinction learning (SCR CS+ > CS−, last three trials, $\beta = 0.21$, SE = 0.10, $T_{30} = 2.19$, $P = 0.04$), which we henceforth used as an index for extinction success. The first model set up to predict CRs at test on day 3 corrected for drug group ($R^2 = 0.17$, $F_{1,33} = 6.58$, $P = 0.02$; $n = 33$). Including extinction success into the model significantly improved the prediction ($\Delta R^2 = 0.11$, $\Delta F_{1,30} = 5.52$, $P = 0.03$). In the subsequent analysis step, we further included potential CS+ offset-related vmPFC reactivations. The combined model explained 40% of the variance ($F_{3,29} = 6.41$, $P = 0.002$) and predicted CRs at test significantly better than extinction success and drug group on their own ($\Delta R^2 = 0.11$, $\Delta F_{1,29} = 5.49$, $P = 0.03$). These results confirm that vmPFC reactivations occurring spontaneously after extinction learning make an independent contribution to explaining long-term extinction memory retrieval, when controlling for interindividual differences in extinction learning.

**Discussion**

In summary, our study shows that spontaneous postextinction reactivations of an activity pattern evoked in the vmPFC by the unexpected omission of the US during extinction learning (day 2) predict long-term extinction memory retrieval (day 3), as assessed by differential SCRs. The relationship between spontaneous CS+ offset-related postextinction reactivations and extinction memory retrieval at test was specific to the vmPFC and could not be detected in any other region of interest (ROI). Furthermore, a higher number of spontaneous vmPFC reactivations was also predictive of greater vmPFC activity at test. In line with previous studies in rodents and humans[24,25,27,28], greater vmPFC activity at test was in turn associated with improved extinction memory retrieval. In the present study, we further manipulated extinction consolidation using L-DOPA. In line with previous work in rodents[17,18] and humans[17], we observed that a postextinction L-DOPA administration enhanced extinction memory retrieval relative to placebo administration. Our findings extend beyond

those reported to date by showing that the effect of L-DOPA on extinction memory retrieval was mediated by the L-DOPA-induced increase in spontaneous postextinction vmPFC reactivations. Thus, pharmacologically elevated dopamine release enhanced extinction consolidation by amplifying spontaneous vmPFC reactivations after learning. This is the first evidence in the human memory literature for a dopamine-dependent amplification of neural pattern reactivations during memory consolidation, as detected in fMRI. Furthermore, these results corroborate that spontaneous postextinction vmPFC reactivations make an independent contribution to extinction memory consolidation.

Fear extinction promotes resilience against the development of post-traumatic stress orders[1] and fear extinction principles are employed during cognitive-behavioral therapy in order to reduce pathological fears[5]. However, in order to prevent the return of fear, the extinction memory has to successfully be retrieved at later confrontations with the CS. Our results show that the long-term retrieval of extinction memories can be explained by interindividual differences in spontaneous postextinction reactivations in the vmPFC. The contribution of spontaneous postlearning reactivations to memory consolidation has intensively been studied in hippocampus-dependent spatial memories[29]. However, in rodents, postlearning reactivations of learning-related activity patterns are also observed in the prefrontal cortex, during sleep as well as awake rest[30–33]. Among others, such prefrontal reactivations were shown to contribute to the memory consolidation of new rules acquired in a rule-shift task[30]. Spontaneous activity in the infralimbic cortex after extinction learning is indispensable for extinction memory consolidation in rodents: when inhibiting NMDA receptor activity in the vmPFC after extinction learning, animals exhibit impaired extinction memory retrieval at test 24 h later[8,9]. Particularly, the number of spontaneous postextinction neuronal bursts in the vmPFC, also observed during extinction[11], predicts later extinction memory retrieval in rodents[8]. Motivated by these findings we focused our analyses on the human vmPFC and provide the first evidence that spontaneous postextinction reactivations of a learning-related vmPFC activity pattern contribute to human extinction memory consolidation and can explain interindividual differences in extinction memory retrieval and vmPFC activity at test.

Our results show that specifically reactivations of the vmPFC activity pattern elicited at the time of the unexpected omission of the US at CS+ offset early in extinction predict extinction memory retrieval. Why may this specific vmPFC pattern be important for extinction memory retrieval? The unexpected omission of the aversive US early in extinction may elicit a surprise, relief or prediction error ("outcome better than expected") signal. It is conceivable that upon such a signal the vmPFC acquires and updates a representation of the expected value of the new "CS–noUS" association, in line with the role of the vmPFC in expected value representation[34]. A limitation of our study is that we did not model the prediction error signal explicitly (due to the limited reliability of model parameters estimated based on noisy trial-by-trial SCR). Thus, future work is necessary to elucidate the contribution of prediction error signals to the formation of the CS+ offset-related vmPFC activity pattern, whose postlearning reactivation supports extinction memory consolidation.

Previous research in humans has revealed that stimulus-specific postlearning reactivations can be identified in the entorhinal and retrosplenial cortices[23], the occipito-temporal cortex[35,36] and the hippocampus[7] using multivoxel pattern analyses in fMRI. Across studies, stimulus-specific postlearning fMRI pattern reactivations were associated with later memory

performance. Our results extend these findings by providing the first evidence for a critical role of dopamine in fMRI pattern reactivations during human memory consolidation. That is, when enhancing extinction memory consolidation pharmacologically by administering the dopamine precursor L-DOPA after learning, we observed an increase in vmPFC reactivations. The L-DOPA induced increase in vmPFC reactivations explained the effect of L-DOPA on CRs at test. Importantly, this finding cannot be explained by an unspecific effect of L-DOPA on detecting any kind of reactivation, as L-DOPA did not affect the number of reactivations of any other control pattern. Our findings are in line with previous research in rodents showing that optogenetic stimulation of dopaminergic projections to the hippocampus during spatial learning amplifies postlearning hippocampal reactivations and results in improved memory performance[21]. Similarly, learning in a reward-context, presumably increasing dopaminergic activation during learning, enhanced fMRI pattern reactivation in humans[7]. Here, we show that also postlearning pharmacological enhancement of dopamine amplifies memory reactivations. Such an amplification of postlearning reactivations is suggested to stabilize synaptic connections[37] and thereby, eventually, result in better memory performance.

The relationship between spontaneous postextinction CS+ offset-related pattern reactivations and extinction memory retrieval was limited to the vmPFC. Control regions, among them the hippocampus, did not show any relation between CS+ offset-related pattern reactivations and later extinction memory retrieval. We can, however, not exclude that reactivations of other patterns than the CS+ offset elicited pattern contribute to extinction memory consolidation in these regions. In light of the reported synchronization of prefrontal activity pattern reactivations and hippocampal sharp wave ripples in rodents[30,38], future studies may also investigate whether the here observed CS+ offset-related vmPFC reactivations are concerted with hippocampal ripple activity.

An important question raised by our results is why vmPFC reactivations specifically 45 min after extinction learning predicted CRs at test, whereas the effect was weaker and non-significant 10 and 90 min after extinction learning. Studies on molecular cascades contributing to synaptic plasticity have shown that different molecular signaling cascades follow different time courses after learning[39], i.e., the disruption of a certain molecular signal does only lead to a disruption in memory consolidation when applied in a sensitive time window. Disruptions preceding or succeeding this signal pathway's specific time window do not affect memory consolidation and performance. Given this temporal specificity on the molecular level, it is possible that also large-scale neural activity pattern reactivations may only be functionally relevant during a specific time window after learning. However, the exact time course of their contribution to memory consolidation has not been investigated yet.

In the present study, we identified spontaneous postextinction reactivations of an extinction learning-related neural pattern in the vmPFC as a basal mechanism of human extinction memory consolidation. By experimentally manipulating dopaminergic activity after learning and observing increases in spontaneous vmPFC reactivation frequency associated with improved extinction memory retrieval, the present study further provides the first evidence for a critical role of dopamine in human memory reactivations. These results open up a wide field of investigation concerning the neurophysiological basis of extinction memory consolidation and its enhancement with pharmacological, behavioral or neurostimulation tools.

## Methods
**Participants**. In total, we recruited 45 male participants. Five participants had to be excluded prior to data analysis due to claustrophobia, leading to an early

termination of the experiment on day 1 ($n = 3$), noncompliance with the instructions ($n = 1$) and a congenital pendular nystagmus ($n = 1$). In total, 40 participants (age range: 26–36 years, mean ± SD: 28.1 ± 2.7) completed the study. We restricted recruitment to male participants as the estrous cycle interacts with extinction memory consolidation[40] and dopamine can have opposing effects on extinction depending on estrous cycle phase[41]. A board-certified physician screened participants for contraindications of L-DOPA intake, current physiological, neurological, or psychiatric disorders, excessive consumption of nicotine (>10 cigarettes/day), alcohol (>15 glasses of beer/wine per week) or cannabis (>1 joint/month), participation in other pharmacological studies, tinnitus (as contra-indication for startle probe exposure). Drug abuse was assessed via urine test (M-10/3DT; Diagnostik Nord, Schwerin, Germany). During the screening session we also tested whether participants showed normal skin conductance responding. To this aim, we attached two electrodes of the eSense Skin response device (Mindfield® Biosystems Ltd., Berlin, Germany) to the medial phalanges of the first and the third finger. Participants were then asked to take several deep breaths. In addition, the physician clapped his hands without announcement inducing a light acoustic startle response. Both deep breathing and acoustic startle usually result in a deflection of the skin conductance, not seen in skin conductance nonresponding individuals. None of the participants screened for the present study had to be excluded according to these criteria. The experiment was approved by the local ethics committee (Ethikkommission der Landesärztekammer, Rhineland-Palatinate, Germany) and conducted in accordance with the Declaration of Helsinki.

**Drug treatment.** Participants were randomly assigned to the L-DOPA or placebo group, with the restriction that groups were matched on ASI[42] and STAI-T[43] questionnaire scores (for comparisons of placebo and L-DOPA group on questionnaire scores, see Supplementary Table 1). Participants were either administered 150/37, 5 mg levodopa-benserazide (Levodopa-Benserazid-ratiopharm®, Germany; for dosage see refs. [17,19]) or an identically looking capsule filled with mannitol and aerosol (i.e., placebo). Drug administration was double-blind. Drugs were prepared and provided by the pharmacy of the University Medical Center Mainz. Participants were asked to refrain from eating, consuming caffeinated drinks, and smoking 2 h prior to drug intake. The short half-life of L-DOPA (~90 min)[26] allowed us to test the effect of L-DOPA on memory consolidation specifically, without risking potential confounding effects of L-DOPA at test 24 h after drug intake.

**Stimuli.** Two black geometric symbols presented in the center of a computer screen served as CS+ and CS− and were super-imposed on background pictures of either a kitchen or a living room, which served as contexts A and B. Assignment of symbols to CS+/CS− and rooms to contexts A/B was counter-balanced between participants and groups. In order to diminish the risk of low visual feature differences between stimuli potentially confounding fMRI BOLD multivoxel patterns, we used a square and a rhombus (square turned by 90°) as CSs and adjusted mean contrast and luminance between pictures using the SHINE toolbox[44]. In all experiments, a painful electrical stimulation consisting of three square-wave pulses of 2 ms (50 ms interstimulus interval) was employed as US. Pain stimuli were generated by a DS7A electrical stimulator (Digitimer, Weybridge) and delivered on the right dorsal hand through a surface electrode with platinum pin (Specialty Developments, Bexley, UK).

**Experimental procedure.** Day 1—Conditioning: Upon arrival participants completed questionnaires on trait and state anxiety (STAIT-T/S)[43], anxiety sensitivity (ASI-3)[42] and demographic data. Electrodes were attached and participants were placed in the MRI scanner. A resting-state scan was conducted before the start of the experiment (Fig. 1a). Subsequently, US-intensity was calibrated to a level rated as "maximally painful, but still tolerable". Familiarization consisted of two CS presentations in both contexts and a practical training of US-expectancy ratings. We instructed participants that the experiment was distributed across 3 days, that one symbol would never be followed by an electrical stimulation, and that their task was to find out what rule applied to the other symbol. Each day, the paradigm started with US-expectancy ratings for each CS (Supplementary Figure 1). Subsequently, the context picture was on screen for 5 s before the first CS was presented. The context remained on the screen continuously throughout the experiment. CSs were presented for 4.5 s. US delivery terminated with CS presentation. Intertrial intervals lasted 17, 18, or 19 s (mean of 18.5 s). Trial order was randomized in such a way that not more than two trials of the same type (i.e., CS +, CS−) succeeded each other. During conditioning in context A participants were presented with ten CS+ and ten CS− trials, and five out of ten CS+ presentations (i.e., 50%) were reinforced with a pain stimulus. After conditioning, participants again rated their US-expectancy. Another resting-state scan was conducted after the conditioning session. Subsequently, electrodes were detached and participants filled out a contingency questionnaire. The total duration of the session amounted to 1.5 h on day 1.

Day 2—Extinction: The extinction session took place approximately 24 h (±2 h) after conditioning. Participants filled out the STAI-S. Before the start of the extinction session electrodes were attached and one resting-state scan was collected. We did not recalibrate the US, but informed participants that their individual US strength from day 1 would be applied again and that the experiment would continue. During the extinction session 15 CS+/CS− trials were presented in

context B. Subsequently, electrodes were detached and participants were administered either a placebo or a L-DOPA pill. Directly after pill intake (~10 min) and 45 and 90 min after pill intake, we conducted one resting-state scan. During breaks participants remained under observation and were provided with magazines. After the last resting-state scan, participants filled out a side-effects questionnaire and the STAI-S. Total session duration amounted to 2.5 h.

Day 3—Test: Approximately 24 h (±2 h) later, participants filled out the side-effects questionnaire and the STAI-S again. After electrode attachment and a resting-state scan, participants were again only instructed that their US strength from day 1 would be applied and that the experiment would continue. During test ten CS+/10 CS− were presented in context B. Total session duration amounted to 1 h.

**Resting-state scans.** We collected 8 min blocks of resting-state fMRI data before and after fear conditioning (day 1), before extinction learning, ~10, 45, and 90 min after extinction learning and drug administration (day 2), and before the test (day 3). During resting-state scans participants were instructed to remain awake, to keep their eyes open and fixate a black cross presented in the center of a gray screen, to let their mind wander freely and to avoid repetitive mental activity[45]. Compliance with the instruction to remain awake was monitored using video recordings of the left eye.

**Skin conductance responses (SCRs).** As in previous work from our laboratory[17,19], differential SCRs to CS+ vs. CS− were used as outcome measures. Electrodermal activity was recorded from the thenar and hypothenar of the nondominant hand using self-adhesive Ag/AgACl electrodes (EL-509, BIOPAC® Systems Inc., Goleta, CA, USA) filled with an isotonic electrolyte medium and the Biopac MP150 with EDA100C device. The raw signal was amplified and low-pass filtered with a cut-off frequency of 1 Hz. Using a custom made analysis script, we manually scored the first local minimum in the skin conductance time course in a window from 900 to 4000 ms after CS onset as response onset. The following local maximum was scored as response peak and SCRs as peak-to-onset difference. Importantly, the experimenter scoring the data was blinded to the stimulus type (CS+/CS−) of each SCR and the group belongingness (placebo/L-DOPA) of each participant. Responses smaller than 0.02 µs were scored as zero and remained in the analysis. If more than 75% of trials had to be scored as zero, data of that subject/day was considered invalid. This applied to $n = 3/4/5$ (day 1/day 2/day 3) participants. For statistical analysis of each day's SCR data we included all available data, i.e., $n = 37/36/35$ on day 1/day 2/day 3, respectively. To correct for interindividual variance, data were log-transformed (+1) and range-corrected[46] within each subject and day.

**US-expectancy ratings.** Before the start and after the end of each experimental phase participants were asked to indicate their expectancy to receive an US for each CS within a 15 s response time window by moving a cursor on a visual analog scale from 0 (= no expectation) to 100 (= very high expectation) by pressing one button. They were asked to confirm their rating with a second button. Button presses were recorded using an MRI compatible response box (LUMItouch, Photon Control Inc., Baxter, Canada). Despite two practice rating trials before the start of the experiment some participants did not confirm their response within the 15 s response time window. Thus, their data were not included into the statistical analysis, resulting in $n = 35/36/38$ on day 1/day 2/day 3 rating data sets for statistical analysis, respectively.

**Statistical analysis.** All statistical analyses were performed in R (https://www.r-project.org/). For statistical analysis of SCR data, we conducted repeated-measures ANOVA with stimulus (CS+/CS−) as within-subject, and group (L-DOPA/placebo) as between-subject factor (R package "car" implementing Type-III Sums of Squares). As single-trial SCRs can easily be affected by factors other than the stimulus-evoked response (e.g., concurrent breathing and a stimulus unrelated movement), we averaged SCRs across several trials to achieve a more reliable measure of the actual stimulus-evoked SCRs. In order to harmonize measures across a series of experiments on the effect of L-DOPA on extinction memory consolidation with varying trial numbers (Gerlicher et al., in preparation), we standardized the operationalization of fear acquisition, fear memory recall, fear at the end of extinction, and extinction memory retrieval a priori across studies. Namely, we operationalized initial fear acquisition as SCRs to CSs averaged across the last 20% of trials on day 1 (here: last two trials), fear memory recall as SCRs to CSs averaged across the first 20% of trials on day 2 (here: first three trials), and fear at the end of extinction as SCRs to CSs averaged across the last 20% of trials (here: last three trials) on day 2. Note, that instead employing the first five and last five trials during extinction, as in the fMRI analysis, does not change the results (Supplementary Table 2). As in previous studies[17,19], the effect of L-DOPA on extinction memory retrieval was tested on SCRs to CS+ and CS−, averaged across the whole test phase on day 3 (i.e., all ten trials). For statistical analysis of US-expectancy rating data, we conducted repeated-measures ANOVA with stimulus (CS+/CS−) as within-, and group (L-DOPA/placebo) as between-subject factor on all pre- and postexperiment US-expectancy ratings. Results were considered significant when below a threshold of $P = 0.05$ (two-sided tests).

**Acquisition of MRI data.** MRI data were acquired on a Siemens MAGNETOM Trio 3 Tesla MRI System using a 32-channel head-coil. Resting-state and task data

for each experimental phase were recorded using gradient echo, echo planar imaging (EPI) with a multiband sequence covering the whole brain (TR: 1000 ms, TE: 29 ms, multiband acceleration factor: 4, voxel size: 2.5 mm isotropic, flip angle 56°, field of view: 210 mm; ref. [47]). A high-resolution T1-weighted image was acquired for anatomical visualization and normalization of the EPI data (TR: 1900 ms, TE: 2540 ms, voxel size: 0.8 mm isotropic, flip angle 9°, field of view: 260 mm). Furthermore, T2-weighted images were collected for preventative neuroradiological diagnostics conducted for all participants (39 slices, TR: 6100 ms, TE: 79 ms, voxel size: 3 mm isotropic, flip angle: 120°). Lastly, we also collected multidimensional diffusion weighted tensor images from each participant (72 slices, voxel size: 2 mm isotropic, TR: 9100 ms, TE: 85 ms, number of directions: 64, diffusion weights: 2, $b$ value 1: 0 s/mm$^2$, $b$ value 2: 1000 s/mm$^2$; data not shown).

**Preprocessing of fMRI data.** fMRI data were preprocessed and analyzed using statistical parametric mapping (SPM12, Wellcome Trust Centre for Neuroimaging, London, UK, http://www.fil.ion.ucl.ac.uk/) running on Matlab 2015b (Math-Works®, Natick, Massachusetts, USA). The first five volumes of each scan were discarded due to equilibrium effects. Preprocessing included realignment and co-registration of functional images to the T1-weighted anatomical image. Subsequently, the T1-weighted anatomical image was segmented and normalized to MNI space based on SPM's tissue probability maps. Normalization of the functional images was achieved by applying the resulting deformation fields to the realigned and co-registered functional images. Lastly, functional data were smoothed using a 6 mm full-width-at-half-maximum Gaussian smoothing kernel.

**Single-subject level analysis of fMRI data during extinction learning (day 2).** In order to investigate potential spontaneous reactivations of CS+ offset-related vmPFC multivoxel patterns, we analyzed the extinction session data (day 2) using a general linear model (GLM), including one regressor each for CS+ onsets, CS− onsets, pre- and postextinction US-expectancy ratings, and context on-/offset. In addition, the model included one regressor for the first five CS+ offsets, at which the omission of the US is unexpected, and one for the first five CS− offsets, at which US omission is expected by the participant. Further, one regressor for the remaining ten CS+ and one regressor for the remaining ten CS− offsets were included. For control analyses (Supplementary Fig. 5B, C) we created an additional single-subject level model including CS+ onsets, CS− onsets, US-expectancy ratings, context on-/offsets and one regressor each for the first, middle, and last five CS+ and CS− offsets, respectively. Including three (first, middle, and last five) instead of two CS offset (first five and last ten) regressors for each CS did not change the results (data not shown). All regressors were modeled as delta-functions and convolved with the canonical hemodynamic response function (HRF). Even though CS onset- and CS offset-related regressors succeeded each other by 4.5 s only, correlations between the HRF-convolved regressors were low (Supplementary Fig. 9). FAST correction for autocorrelation and a high-pass filter (128 s) were applied.

**Multivariate fMRI analysis of postextinction pattern reactivation.** The multivoxel patterns evoked by the first five US omissions at CS+ and CS− offset in the vmPFC region of interest (ROI; Harvard-Oxford Atlas, Harvard Center for Morphometric Analysis, thresholded at 50% tissue probability) were extracted from the resulting beta-maps in the vmPFC. Resting-state data were analyzed in accordance with a previous study examining associative memory reactivation[23], i.e., GLMs for each resting-state scan (pre- and 0, 45, and 90 min postextinction) included delta-function regressors for each volume, thereby accounting for potential reactivations, which may occur during any point of the resting-state scan. No high-pass filtering was applied in the resting-state models, and AR(1) autocorrelation correction was employed. Multivoxel patterns during the resting-state were extracted from the resulting beta-image series in the vmPFC (TR: 1 s, i.e., $480 - 5 = 475$ beta images). Subsequently, we correlated (Pearson correlation coefficient) the patterns evoked by the first 5 CS+ and CS− offsets with the resulting 475 patterns of all four resting-state scans and Fisher Z-transformed the correlation coefficients. The 475 correlations of the CS+/CS− offset patterns with the resting-state patterns recorded before extinction learning were employed to create a baseline distribution. The mean and standard deviation of this baseline distribution were used to transform each correlation between the CS+ and CS− offset pattern and the *postextinction* resting-state patterns into a Z-score ($Z_i = (r_i - \mu)/\sigma$), following a previous study[23]. In line with previous work[23], correlations with a Z-score exceeding a value of 2 ($Z > 2 \approx P < 0.05$) were considered potential spontaneous postextinction reactivations of the CS+/CS− offset pattern. Note, however, that results do not depend on the exact threshold employed for defining potential reactivations (see Supplementary Fig. 3 for results with more liberal or conservative thresholds). Reactivations were summed per participant and resting-state scan. The resulting reactivation numbers were log-transformed before analysis in order to account for non-normality of the distribution (Kolmogorov–Smirnov test before transformation: all $Ds(40) \geq 0.17$, all $Ps \leq 0.004$; after transformation: all $Ds(40) \geq 0.13$, all $Ps \leq 0.08$; but see Supplementary Fig. 2 for results with nontransformed data). Subsequently, multiple linear regression analysis with the (log-transformed) number of potential CS+ and CS− offset vmPFC pattern reactivations at all resting-states scans (day 2) as independent and CRs (mean SCR CS+ > CS−) at test (day 3) as dependent variable was

performed (main text Fig. 2b). The effect of L-DOPA on the number of potential vmPFC reactivations was assessed using repeated-measures ANOVA with time (pre- and 10, 45, 90, postextinction) as within- and group (placebo/L-DOPA) as between-subject factor (main text Fig. 2e, f). In addition to our a priori hypothesis concerning the vmPFC, we also tested the relationship between CRs at test (mean SCR CS+ > CS−, day 3) and postextinction number of reactivations in the vmPFC and eight further anatomically defined control ROIs to assess regional specificity of the vmPFC findings using multiple linear regression (Supplementary Fig. 4). Following previous work[48], the following anatomical ROIs were employed due to their involvement in fear and extinction learning based on the Harvard-Oxford Atlas (Harvard Center for Morphometric Analysis, thresholded at 50% tissue probability): bilateral amygdala, hippocampus and insula, as well as the superior frontal gyrus (SFG) and anterior cingulate cortex (ACC). Results were considered significant when below $P = 0.05$ (two-sided tests).

**Single-subject level analysis of fMRI data during test (day 3).** In order to investigate the effect of number of spontaneous postextinction vmPFC reactivations (day 2) on fear-related (CS+ > CS−) BOLD responses at test on day 3 (main text Fig. 2c), we analyzed the preprocessed fMRI data using a GLM with one regressor for CS+ and CS− presentations each, pre- and post-test US-expectancy ratings and context on-/offsets. All regressors were modeled as delta-functions and convolved with the HRF. FAST correction for autocorrelation and a high-pass filter (128 s) were applied.

**Group level analysis of univariate fMRI data.** In order to test the effect of number of spontaneous postextinction vmPFC reactivations (day 2) on fear-related (CS+ > CS−) BOLD responses at test on day 3 on the group level (main text Fig. 2c), we employed SPM's multiple regression with the number of potential vmPFC pattern reactivations, group and their interaction as regressors and the CS+ > CS− contrast at test on day 3 as outcome measure. Results were considered significant when exceeding a threshold of $P = 0.05$, small-volume and family-wise error corrected in the ROIs specified for multivariate main- and control analyses[48], i.e., bilateral amygdala, hippocampus and insula, as well as SFG, ACC, and vmPFC.

**Mediation analysis.** To test whether the effect of L-DOPA on CRs at test on day 3 was potentially mediated by the number of potential CS+ offset-related vmPFC reactivations (45 min postextinction), we conducted mediation analysis (R package "mediation"). Results (main text Fig. 2g) were considered significant when below $P = 0.05$ (two-sided tests).

## Data availability
The data that support the findings of this study are available upon request from the corresponding author A.M.V.G. (e-mail: a.m.v.gerlicher@uva.nl).

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

## Acknowledgments

We thank A. Schick, M. Ilhan, J. Behr, P. Seifert, N. Schappe, K. Yuen and A. Droby for support with data collection and G. Fernández, E. Hermans and B. Meyer for helpful comments on an earlier version of the manuscript. This work was funded by the Deutsche Forschungsgemeinschaft (CRC1193; subproject C01 to R.K., subproject C04 to O.T.).

## Author contributions

A.M.V.G., O.T. and R.K. designed the experiment; A.M.V.G. and O.T. collected the data; A.M.V.G., O.T. and R.K. analyzed the data; and A.M.V.G. and R.K. wrote the paper.

## Additional information

**Competing interests:** The authors declare no competing interests.

