## [Peer Review File · Nature Communications]

Reviewers' comments:

Reviewer #1 (Remarks to the Author):

In this manuscript, Gerlicher and colleagues report a multi-session fMRI study on the effects of dopamine-mediated offline reactivation on extinction learning in a fear conditioning paradigm. Participants were first trained to associate a geometric shape (CS+) with an electrical shock (US) (Day 1). On Day 2, extinction learning took place, followed by the administration of L-Dopa or Placebo. Fear responses were finally tested on Day 3. The authors observed that extinction learning patterns in vmPFC were spontaneously reinstated during post-learning rest periods, with the amount of reactivation predicting extinction success (as measured via skin conductance) on Day 3. Moreover, L-Dopa further increased the amount of reactivations and bolstered extinction learning, pointing to a causal relationship between dopamine, memory reactivation in vmPFC and consolidation.

The manuscript is well written and the research question is timely and relevant. The paradigm is elegant in its simplicity and all analyses are straight-forward and convincing. Effects are robust, and I find the control analyses, using different thresholds for detection reactivations and using different transformations, particularly compelling. The only somewhat arbitrary decision is to consider the first 5 trials during extinction learning as the critical time period. Is there a motivation in the behavioural patterns or in the literature for this choice? Alternatively, can the authors also show that the results don't hinge on the exact choice of initial trials included?

Reviewer #3 (Remarks to the Author):

In the reported fMRI pavlovian fear conditioning study, the authors address two questions: (1) whether, within VMPFC, cross-voxel activation patterns associated with CS presentation and US omission during early extinction can be observed during later resting state scans (in line with consolidation of these representations) and whether the number of such 'reactivations' inversely predicts the strength of conditioned responses at test (extinction memory recall) the day after, and (2) whether administration of L-DOPA after extinction training increases the number of CS-no US reactivations in VMPFC and if this mediates the effect of L-DOPA on extinction memory (i.e. reduced conditioned responses) at test.

The authors give a comprehensive and valuable, if somewhat dense, introduction. They detail how they build on the rodent literature where number of infralimbic bursts during consolidation of extinction training predicts extinction memory retrieval at test 24 hours later. To translate this into a paradigm suitable for fMRI with human participants, the authors take advantage of methodology developed by Staresina et al. 2013. The authors of this prior paper showed how multivoxel pattern analysis could be used to compare activation patterns at encoding with activation patterns during each volume of a later resting state scan. They further showed that re-activation of encoding patterns was stronger for items later recalled than for items later forgotten. The current authors have nicely drawn on this clever paradigm to address the question of whether the representations encoded during extinction training (namely the pairing of the CS+ with the non-occurrence of the US) are similarly 'replayed' during rest and if the extent of such replay can predict extinction memory and explain the influence of LDOPA upon extinction memory. The study is hence both theoretically well motivated and methodologically well designed (though see comment 1 on modeling below). The results are compelling and the discussion well written. I believe this paper will make a valuable contribution to the field.

Comments

Major

1) Regressors used in modeling the extinction learning data.

The authors used the following regressors in their GLM: " one regressor for CS+ onsets, CS-

onsets, pre- and post-extinction US expectancy ratings and context on-/offset each. In addition, the model included one regressor for the first 5 CS+ offsets, at which the omission of the US is unexpected, and the first 5 CS- offsets, at which US omission is expected by the participant. In addition, one regressor for the remaining 10 CS+ and one regressor for the remaining 10 CS- offsets were included. For control analyses (Supplementary Fig. 5B and C) we created an additional single-subject level model including CS+ onsets, CS- onsets, US expectancy ratings, context on-/offsets and one regressor each for the first, middle and last 5 CS+ and CS- offsets, respectively."

I am concerned that given the relatively short fixed duration of the CS+ (4.5s) there is likely to be substantial collinearity between the CS+ onset regressor and the 2 CS+ offset regressors in the main model (after convolution with the hrf), similarly for the CS- onset regressor and the 2 CS- offset regressors. The same goes for the control analysis model. Please can the authors give the post-convolution correlations between regressors in a supplementary figure. If there is indeed high correlation (and hence shared variance) between the CS onset and offset regressors, the authors need to address how this impacts their analysis.

2) My second comment/concern also relates to the authors' estimation of the CS no US activation pattern at encoding. The authors chose to average activation at CS+ offset across the first five (of 15) trials where the CS+ was presented without the US. They justify this as follows:

"Early in the extinction session, US omission is unexpected to the participants and elicits a prediction error. Prediction errors drive the correction of the US prediction associated with the CS and induce extinction learning. To capture this time point, we extracted the average multivoxel vmPFC activity pattern evoked by the first 5 offsets of the CS+ (the CS that had previously been paired with the US during conditioning) during extinction.)"

The authors return to this in the discussion to argue that "it is conceivable that a prediction error signal elicited by the unexpected omission of the US in the mesostriatal dopamine system early in extinction learning induces the creation of a representation of the CS-noUS association or a representation of the latent 'extinction cause' in the vmPFC, which is then reactivated during extinction memory consolidation."

Here the authors are arguing that the early trials are key because these are where the prediction error will be strongest and it is this error which induces the CS-no US representation which is later reactivated. However averaging across the first five trials is a fairly crude way to capture the extent of the prediction error on each trial. This would be easily modeled, allowing for the pattern on each CS+ no US trial to be weighted accordingly. This is unlikely to dramatically change the results but would better fit the current discussion/framing of the results as given above.

3) An initial concern of mine, while reading the paper, was that a few methodological choices seemed fairly arbitrary and I worried whether the results obtained would have been robust to different choices. For example, the authors use the first 5 trials of extinction to represent the early phase of extinction learning for the BOLD activation patterns (and the last 5 to represent late extinction training) but only the first / last 3 trials for the skin conductance data. It would be helpful if the authors could show what would have been obtained if the first (and last) 5 trials had been used for the skin conductance analyses too. If they feel it is important to only use 3 trials a justification for this should be given. Similarly, why did the authors use all trials at extinction memory test?

I note that in other places where decisions might have seemed arbitrary, the authors have done a great job in the supplementary figures of showing how the decisions taken did not impact the results obtained. This went a long way to strengthening my faith in the results.

4) Figure 2c and Fig S2, S3 – please do not use uncorrected thresholds to show small volume activations – please use the corrected thresholds as in the main text. Otherwise the activations displayed are misleading.

Minor

- 1) Please refer to last 3 trials not last 20% of trials – otherwise the reader has to pause, go to the methods to find the number of trials and figure out how many 20% equates to – it puts unnecessary work on the reader

- 2) 'Fully mediated' and 'complete mediation' implies the entire relationship can be explained by (i.e. 100% of variance); the authors should be careful in using this term as opposed to 'significantly mediated'

- 3) Abstract line 1 – should be 'have' not 'has'

- 4) Abstract last line: "Hence, a spontaneous dopamine-dependent memory consolidation-based mechanism underlies the long-term behavioral effects of fear extinction." The results suggest this might be the case but do not categorically prove it – I find this phrasing too definitive, I'd soften it a bit 'might underlie'.

- 5) Pg 10, line 9 'Spontaneous vmPFC activity' – do you mean infralimbic?

- 6). Please can you elaborate on how participants were excluded for 'skin conductance non responding'

- 7) Similarly, please elaborate on how 'first response onset' is defined for:
"SCRs were scored offline as the difference between first response onset in a time window from 900 to 4000 ms after CS onset and the subsequent peak, using a custom-made analysis script."

8. "In order to account for physiological noise and harmonize measures across the different experimental phases, we operationalized initial fear acquisition as SCRs to CSs averaged across the last 20% of trials on day 1..."
How does this account for physiological noise? Please explain.

Reviewer #1 (Remarks to the Author):

In this manuscript, Gerlicher and colleagues report a multi-session fMRI study on the effects of dopamine-mediated offline reactivation on extinction learning in a fear conditioning paradigm. Participants were first trained to associate a geometric shape (CS+) with an electrical shock (US) (Day 1). On Day 2, extinction learning took place, followed by the administration of L-Dopa or Placebo. Fear responses were finally tested on Day 3. The authors observed that extinction learning patterns in vmPFC were spontaneously reinstated during post-learning rest periods, with the amount of reactivation predicting extinction success (as measured via skin conductance) on Day 3. Moreover, L-Dopa further increased the amount of reactivations and bolstered extinction learning, pointing to a causal relationship between dopamine, memory reactivation in vmPFC and consolidation.

The manuscript is well written and the research question is timely and relevant. The paradigm is elegant in its simplicity and all analyses are straight-forward and convincing. Effects are robust, and I find the control analyses, using different thresholds for detection reactivations and using different transformations, particularly compelling.

The only somewhat arbitrary decision is to consider the first 5 trials during extinction learning as the critical time period. Is there a motivation in the behavioural patterns or in the literature for this choice? Alternatively, can the authors also show that the results don't hinge on the exact choice of initial trials included?

Reply: We want to thank the reviewer for her/his supportive comments on our study and for raising this important concern. It is indeed correct that the decision to consider the first 5 trials was taken rather arbitrarily. That is, it was merely based on the motivation to extract CS+ offset-related activity patterns from the *early* phase of extinction learning in an extinction paradigm with a total number of 15 trials. Importantly, however, the results remain predominantly robust when instead including the first 3, 4, 6, 7 or 8 trials (see Figure below). In contrast, including less than 3 or more than 8 trials does not yield any significant relation to differential conditioned responses (CRs) or vmPFC activity at test. Also, including trials from the middle or end of the extinction session only (e.g. the middle 5 or last 5 trials) does not yield the same results (Supplementary Figure 5b and c). Together these results indicate that it is critical that the regressors used for estimating the CS+ offset-related activity pattern cover trials in which the surprise about the omission of the US is still high. We have now included these control analyses into the

supplementary information and refer to these results in the main text (p. 7, l. 28) of the revised manuscript. We hope we could thereby address the reviewer's concern adequately.

Supplementary Figure 6. | The effects are stable against changes in the number of trials employed for estimating the CS+ offset-related vmPFC activity pattern from early extinction. For the main analysis we selected the vmPFC activity pattern evoked by the first 5 CS+ offsets early during extinction. To test the robustness of our results against changes in the exact number of trials used for estimating the CS+ offset-related vmPFC activity pattern, we repeated all analyses based on the first 3 to 8 CS+ offsets. The prediction of CR at test (SCR CS+>CS-) on day 3 was not significant when defining the vmPFC activity pattern based on **a)** the first 3 CS+ offsets ($\beta=-.05$, $P=.14$; $n=35$). However, the prediction of CR at test (SCR CS+>CS-) on day 3 remained robust when defining the vmPFC activity pattern based on **b)** the first 4 CS+ offsets ($\beta=-.08$, $P=.02$; $n=35$), **c)** the first 6 CS+ offsets ($\beta=-.09$, $P=.005$; $n=35$), **d)** the first 7 CS+ offsets ($\beta=-.07$, $P=.009$; $n=35$) and **e)** the first 8 CS+ offsets ($\beta=-.07$, $P=.02$; $n=35$). Including drug and its interaction with CS+ offset related vmPFC reactivations into the regression model did not change the results and did not reveal any difference in the relation between vmPFC reactivations and CR at test between placebo and L-DOPA treated participants, as in the main analysis (all $P_s>.42$). Recruitment of the vmPFC during test (CS+>CS-) could also be predicted based on spontaneous reactivations of **f)** the first 3 CS+ offsets ($x,y,z=6,46,-14$; $Z=4.02$, $P=.006$, SVC, FWE; $n=40$), **g)** the first 4 CS+ offsets ($x,y,z=4,34,-20$; $Z=3.61$, $P=.02$, SVC, FWE; $n=40$), **h)** the first 6 CS+ offsets ($x,y,z=6,46,-14$; $Z=5.34$, $P<.001$, SVC, FWE; $n=40$), and **i)** the first 7 CS+ offsets ($x,y,z=8,46,-14$; $Z=3.90$, $P=.009$, SVC, FWE; $n=40$), but the effect was only trend-wise significant for **j)** the first 8 CS+ offsets ($x,y,z=6,44,-14$; $Z=3.27$, $P=.061$, SVC, FWE; $n=40$). Display threshold $P<.001$, uncorr., no masking applied. **k)** There was no significant effect of L-DOPA on number of vmPFC activity pattern reactivations computed based on the first 3 CS+ offsets (repeated measures ANOVA: time x drug $F_{3,114}=.48$, $P=.70$; $n=40$). **l)** There was a trend-wise significant effect of L-DOPA on the number of vmPFC activity pattern reactivations computed based on the first 4 CS+ offsets (repeated measures ANOVA: time x drug $F_{3,114}=2.30$, $P=.08$; $n=40$) specifically 45 min after extinction (post-hoc two-sample t tests: pre: $P=.12$; post: 10 min: $P=.34$; 45 min: $T_{38}=-1.95$, $P=.06$; 90 min: $T_{38}=-1.86$, $P=.07$; $n=40$). **m)** There was a trend-wise significant main effect of drug independent of time on the first 6 CS+ offsets (repeated measures ANOVA: drug $F_{1,38}=3.66$, $P=.06$; $n=40$) due to significantly more vmPFC reactivations 45 min after extinction in L-DOPA treated participants (post-hoc two-sample t tests: pre: $P=.42$; post: 10 min: $P=.20$; 45 min: $T_{38}=-2.09$, $P=.04$, Cohen's $d=.67$; 90 min: $T_{38}=-1.46$, $P=.15$; $n=40$). **n)** The number of reactivations of the first 7 CS+ offset vmPFC activity pattern was significantly greater in the L-DOPA group (repeated measures ANOVA: drug $F_{1,38}=4.39$, $P=.04$, partial $\eta^2=.07$; $n=40$), due to an effect of L-DOPA on number of vmPFC reactivations 45 min after extinction (post-hoc two-sample t tests: pre: $P=.78$; post: 10 min: $P=.50$; 45 min: $T_{38}=-2.32$, $P=.03$, Cohen's $d=.71$; 90 min: $T_{38}=-1.90$, $P=.06$; $n=40$). **o)** Lastly, on the first 8 CS+ offsets there was a trend-wise significant time by drug interaction (repeated measures ANOVA: time x drug $F_{3,114}=3.10$, $P=.06$, Greenhouse-Geisser corrected) with L-DOPA treated participants showing significantly more vmPFC reactivations specifically 90 min after extinction (post-hoc two-sample t tests: pre: $P=.39$; post: 10 min: $P=.73$; 45 min: $P=.12$; 90 min: $T_{38}=-3.11$, $P=.004$, Cohen's $d=.97$; $n=40$). Note, that the effects are not robust to repeating the analyses with vmPFC activity patterns evoked by less than 3 or more than 8 CS+ offsets.

Page 7, line 28: [...] Even though the relation was specific to the pattern elicited by the unexpected US omission at CS+ offset early during extinction, the exact number of trials used for estimating the early CS+ offset-related vmPFC activity pattern (here: first 5 CS+ offsets) was not critical for the results (Supplementary Figure 6).

Reviewer #2 (Remarks to the Author):

The authors report a between-subjects double-blind placebo-controlled fMRI experiment in humans. They find that the administration of L-dopa following extinction training reduces threat conditioned responses 24h later. The administration of L-dopa enhances the reoccurrence of a neural signal that reflects extinction learning during a post-learning resting state scan, which mediates the reduction in threat responses 24h later. These results suggest that increasing dopamine signaling enhances reactivation of an extinction memory trace to enhance memory consolidation and reduce subsequent return of threat (“fear”) responses.

Overall I am excited about this work, found the manuscript clear and a pleasure to read, and am convinced it will appeal to a broad audience. I do have two major (related) concerns, some minor concerns, and a few suggestions. If the authors can address these concerns I would be more than happy to support publication.

Reply: We thank the reviewer for the supportive comments on our study and the thorough and thoughtful review.

Major concerns:

The authors conclude that: “In summary, our study shows that spontaneous post-extinction reactivations of an activity pattern evoked in the vmPFC by the unexpected omission of the US during extinction learning (day 2) predict long-term extinction memory retrieval (day 3), as assessed by differential skin conductance responses”. But I am not fully convinced that the data unequivocally supports this conclusion.

- 1. Although the authors report that L-dopa diminishes the spontaneous recovery of conditioned responses, on the first trial of the test the difference in responses between the CS+ and CS- in the L-dopa group actually seems to be greater than in the placebo group. Moreover, eye-balling the data in Figure 1, the mean difference in responses across all test trials appears to be driven by more rapid extinction learning during test in the L-dopa group. This raises the possibility that L-dopa does not reduce spontaneous recovery but primes more rapid subsequent extinction learning during test. This would drastically change the interpretation of the results and its clinical implications.*

Reply: We thank the reviewer for raising this important point. As indicated by the reviewer, the CS+ evoked SCR on the first test trial is indeed numerically greater in the L-DOPA (mean SCR CS+: .65) than the placebo (mean SCR CS+: .54) treated participants. However, two-sample *T*-tests on CS evoked SCRs on the first test trial on day 3 did not reveal any significant difference between groups on CS+ ($T_{33}=1.19$, $P=.24$) or CS- evoked responses ($T_{33}=-.18$, $P=.86$) or their difference (CS+>CS-; $T_{33}=.98$, $P=.34$). Further, even though the graph of SCRs on day 3 seems to suggest that differential conditioned responses (CRs) in the L-DOPA group extinguished faster across the test session on day 3, a repeated measures ANOVA with group (placebo/L-DOPA) as between-subject factor and stimulus (CS+/CS-) and trial (trial 1-10) as within-subject factors indicated that the time course of re-extinction during test on day 3 does not differ between groups. That is, the group x stimulus effect remained significant ($F_{1,297}=6.57$, $P=.02$, partial $\eta^2=.17$), whereas the group x stimulus x trial interaction was not significant ($F_{1,297}=1.32$,

$P=.23$). Exploratory post-hoc analyses did also not reveal any significant linear ($F_{1,297}=.86$, $P=.36$) and a merely trend-wise significant quadratic ($F_{1,297}=3.94$, $P=.06$) group x stimulus x trial effect. Together these results do not provide statistical support for a faster re-extinction during test after post-extinction L-DOPA administration. They rather indicate that L-DOPA resulted in a general reduction of differential CRs across test on day 3. However, we fully agree with the reviewer that the question of whether L-DOPA might have sped up re-extinction during test on day 3 arises automatically when looking at the SCRs presented in Figure 1 during test on day 3. We therefore now provide the reader with the latter results in the legend of Figure 1 (see below).

Generally, when interpreting the results it is important to consider that L-DOPA has a short half-life of only 90 minutes (Contin and Martinelli, 2010; LeWitt, 2015; Nyholm *et al*, 2012). This allows us to exclude that the post-extinction L-DOPA administration on day 2 had any direct effect on re-extinction learning during test on day 3. Instead, post-extinction L-DOPA administration on day 2 may have only affected CRs at test on day 3 indirectly, by effects on extinction memory consolidation processes on day 2. Thus, we are confident that the L-DOPA effects are truly memory related and are not a reflection of any drug-induced state at test on day 3.

Page 14:

Figure 1 | Experimental design and skin conductance responses. [...] The groups differed significantly on mean SCRs across the test phase on day 3 (marked by yellow line) due to significantly smaller mean SCRs to the CS+ in L-DOPA- compared to placebo-treated participants. Note, that there was only a significant difference between groups on mean SCRs across the test phase, but the speed of re-extinction did not differ significantly between groups (control analysis with stimulus (CS+, CS-) and trial (1-10) as within-, and group (placebo, L-DOPA) as between-subject factor: stimulus x group, $F_{1,297}=6.57$, $P=.02$, partial $\eta^2=.17$; stimulus x trial x group, $F_{1,297}=1.32$, $P=.23$; $n=35$).

In addition, this would require confirming that the “reactivations” in the neural data that are reported are truly learning and memory related and not some reflection of state effects. This would require a different analytical approach to the fMRI data (see the next point).

- 2. An important concern for resting state “memory reactivation” fMRI analyses, and particular in the case of fear conditioning experiments, is that the detected correlations are not driven by non-cognitive effects such as movement, heart-rate and other physiological responses that may be linked to stimulus conditions. This may be particularly problematic if drug administration biased such effects. Previous studies have taken efforts to measure and regress out physiological responses and have, for example, used white matter and CSF signal to correct for movement.*

Reply: We had oriented the reactivation analysis of the resting-state data closely on the procedure reported by Staresina, Alink, Kriegeskorte & Henson (2013), who had not applied any such correction. We do, however, fully agree with the reviewer that it is important to exclude that physiological confounds affected the results. To this aim, we repeated the analysis after regressing out nuisance signals, as mentioned by the reviewer. Namely, six head motion parameters, mean white matter and mean CSF signals were removed from each resting-state

time course using the “regress out covariates” function provided by the Resting-State fMRI Data Analysis Toolkit, REST (Song *et al*, 2011). Subsequently, we repeated the main analyses on the nuisance-adjusted resting-state time courses. Importantly, the results do not change after removal of the nuisance signals (see Supplementary Figure 7 below). We now point the reader to this additional analysis in the main text (p. 7, l. 28) and provide the following results in the supplementary.

Supplementary Figure 7 | Regressing out nuisance signals from the resting-state time courses before the reactivation analysis does not change the results.

The resting-state data were analyzed in accordance with a previous study (Staresina *et al*, 2013) that did not explicitly control for the influence of spontaneous fluctuations in physiological signals of no interest or head motion. In order to test whether our results were affected by such nuisance signals we repeated the analysis on resting-state time courses cleared from mean cerebrospinal fluid, mean white matter and the six head motion signals using the ‘regress out covariate’ function provided by the Resting-State fMRI Data Analysis Toolkit (Song *et al*, 2011) (REST). **(a)** The prediction of CR at test on day 3 based on potential spontaneous CS+ offset-related vmPFC reactivations (45 min) after extinction remained robust (linear regression: $\beta = -.11$, $P = .001$; $n = 35$). Including drug and its interaction with the number of reactivations into the regression model did not change the effect (multiple linear regression: $\beta_{\text{react}} = -.10$; $P = .03$) and indicated that the effect did not differ significantly between groups ($\beta_{\text{drug} \times \text{react}} = -.03$, $P = .69$; $n = 35$). **(b)** Similarly, the prediction of vmPFC activity (CS+>CS-) during test on day 3 based on potential vmPFC pattern reactivations remained robust (SPM multiple linear regression: $x, y, z = 6, 48, -12$; $Z = 3.84$, $P = .01$; SVC, FWE; $n = 40$). Display threshold $P < .001$, uncorr., no masking applied. **(c)** There was a trend-wise significant effect of drug on number of CS+ offset-related vmPFC reactivations (repeated measures ANOVA: drug $F_{1,38} = 3.99$, $P = .05$, partial $\eta^2 = .10$; $n = 40$), due to significantly greater number of reactivations in L-DOPA compared to placebo treated participants 45 min after extinction (two-sample T -test: post-hoc t tests: pre: $P = .66$; post: 10 min: $P = .17$; 45 min: $T_{36} = -2.42$, $P = .02$, Cohen’s $d = .77$; 90 min: $P = .40$; $n = 40$). In addition, the mediation of the effect of L-DOPA on CR at test by the number of potential spontaneous CS+ offset-related vmPFC pattern reactivations remained robust (indirect mediation effect: $\beta = -.06$, 95% CI: $-.11$ – $-.02$, $P = .01$; $n = 35$).

Page 7, line 28: Lastly, the results are not dependent on clearing or not-clearing the resting-state BOLD activity time courses from nuisance signals (i.e. cerebrospinal fluid, white matter, head motion) before identifying potential vmPFC reactivations (Supplementary Figure 7).

Moreover, and potentially more important, a concerns for all memory reactivation research is to ensure that the detected neural “reactivations” truly reflect learning and memory representation and not some non-specific effects, state-effects, or pre-existing

correlational structures. For this reason, previous research has often used methods to establish that neural activation patterns post-learning correlate with learning related patterns over and beyond pre-extinction patterns for example by using partial correlations. The authors do show more learning related activation patterns after compared to before learning, but this may simply be the result of differences in MR scanner functioning between consecutive days. Why have the authors not used such measures and can they exclude that their reactivation measure is not biased by such non-learning and -memory related confounds?

Reply: As mentioned by the reviewer, previous studies (e.g. Hermans *et al*, 2016) computed the partial correlation coefficient between a multi-voxel correlation structure from the learning phase and a correlation structure from a post-learning resting-state while controlling for variance explained by a pre-learning resting-state correlation structure within a ROI. Note, that in these analyses the amount of explained variance (vs. reverse explained variance) was used to determine whether a learning-related correlation structure explained significant variance in a post-learning correlation structure. In contrast, in the analysis as introduced by Staresina, Link, Kriegeskorte & Henson (Staresina *et al*, 2013) and applied here, one uses the correlation coefficients between a stimulus-specific activity pattern from learning and each pre-learning activity pattern during a baseline resting-state scan in order to threshold the post-learning correlation coefficients between a stimulus-specific activity pattern from learning and each post-learning activity pattern. Thus, this analysis only classifies a post-extinction correlation coefficient as a potential reactivation if it exceeds a threshold determined by pre-learning baseline correlation coefficients. Thereby, also the analysis applied here controls for pre-extinction pattern similarities. Note, that pre- and post-extinction resting-state scans were collected on a single day (day 2) such as that differences in MR scanner functioning between consecutive days cannot account for the effects.

In addition, our main regression model (main text, page 6) again controls for variance explained by pre-extinction supra-threshold pattern similarities by predicting CR at test based on number of potential vmPFC reactivations during the pre-extinction baseline and all post-extinction resting-state scans (multiple linear regression model: CR at test ~ baseline react + post10 react + post 45 react + post 90 react). That is, the main result reported on page 6 of a relation between reactivations 45 min post-extinction and CR at test (multiple linear regression: $\beta_{45\text{min}} = -.13$, $SE = .03$, $T_{30} = -4.05$, $P = .0003$; $n = 35$) is controlled for by the number of baseline, post 10 and post 90 min vmPFC reactivations and the result is roughly equivalent to conducting a partial correlation analysis between CR at test and vmPFC reactivations 45 min post-extinction, while controlling for pre-extinction pattern reactivations (partial $R = -.58$, $P = .0003$, $n = 35$; R package 'ppcor', using the variance-covariance matrix to compute the partial correlation coefficient).

Lastly, in contrast to previous studies our experimental design allows us to exclude that the reactivation measure is non-learning/memory related. The present study included an experimental manipulation of memory consolidation, namely a post-extinction placebo/L-DOPA administration. Notably, the administration of L-DOPA specifically increased the number CS+ offset-related vmPFC reactivations, whereas it left the number of reactivations of any of the tested control vmPFC activity patterns unaffected (main text Figure 2f and Supplementary Figure

5). Together, these results strengthened our conviction that the reported effects cannot be attributed to non-learning or –memory related confounds. We hope that they also address the reviewer’s concerns adequately.

Minor concerns

1. *It wasn’t entirely clear to me if this is a re-analyses of the same data set as used in the Haaker 2013 PNAS paper (I suspect it is not)? If so this should be acknowledged. If not it can be made clearer up front. If it is the same data set then the behavioral results would not be novel (again, I don’t think this is the case) and if it is a new data set then please stress more that this is a replication (something much needed in neuroscience!). In addition, it should be made clear that the Haaker paper already reported that changes in vmPFC-amygdala connectivity and amygdala activity are related to reduced SCR during a renewal test. This would place the current findings on a stronger foundation*

Reply: We apologize for the lack of clarity. We want to emphasize that the data for the present manuscript has not been published before. Instead, the data was collected with the designated purpose of investigating spontaneous post-extinction memory consolidation processes. To this aim, we employed the following changes in comparison to the paradigm reported in Haaker et al. (Haaker et al, 2013):

	Haaker et al. (2013)	Present study
Paradigm:	Day 1: Condit. (A), Extinct. (B), Condit. (A), Extinct. (B) Day 2: Test (A,B,A,B,A,B,....)	Day 1: Conditioning (A) Day 2: Extinction (B) Day 3: Test (B)
L-DOPA administration:	Day 1: immediately after fear conditioning and extinction	Day 2: immediately after extinction
Effect of L-DOPA on:	ABA renewal	ABB extinction retrieval (i.e., spontaneous recovery)
Stimulus duration:	3 sec	4 sec
ITI duration:	3.6 sec (2.5-7 sec)	18.5 sec (17-20 sec)
Contextual stimuli:	colored background	rooms
Distractor task:	“Maintain attention”	none

Critically, by distributing conditioning and extinction on two consecutive days in the present study instead of conducting conditioning and extinction on day 1 (Haaker et al., 2013), the effect of L-DOPA was confined to the post-extinction memory consolidation phase on day 2. In addition, pain stimulus delivery, fear learning itself and fear memory consolidation occurred on day 1 and

did not affect extinction learning or the post-extinction memory consolidation phase on day 2. Hence, in contrast to the Haaker study, we can here exclude any L-DOPA effect on the immediate consolidation of the fear conditioning memory. Furthermore, the present study did not exactly replicate the effect of reduced ABA renewal after L-DOPA as reported in Haaker et al. (2013), as we here specifically tested the effect of L-DOPA on long-term extinction retrieval (i.e., spontaneous recovery). Hence, the present study does reconfirm an effect of L-DOPA on extinction consolidation, but at the same time can at best be labeled a “conceptual replication” of Haaker et al. (2013). We now emphasize the confirmatory nature of our results more strongly and differentiate them from previous work in the discussion on p. 9, l. 29:

Page 9, line 29: Confirming previous work in rodents^{17,18} and humans¹⁷ we observed that a post-extinction L-DOPA administration enhanced extinction memory retrieval relative to placebo administration. In addition to these previous studies, we can now show that the effect of L-DOPA on extinction memory retrieval was mediated by the L-DOPA-induced increase in spontaneous post-extinction vmPFC reactivations.

It does, however, raise interesting questions. First, is the number of detected reactivations during post-extinction rest related to vmPFC-amygdala connectivity during test and not just to vmPFC activity? .

Reply: We agree with the reviewer that this is an interesting follow-up question. In order to test this question, we employed general psychophysiological interaction analysis (gPPI; McLaren et al., 2012) on the fMRI data of the test session on day 3. The single-subject GLM included CS+, CS- (‘psychological factors’), rating and context on-offset regressors as well as the eigenvariate of the timecourse of the vmPFC seed region (‘physiological factor’) and its interaction with the CS+ and CS- presentations, respectively (‘psycho-physiological factors’). First-level contrasts (CS+ x vmPFC > CS- x vmPFC) were computed on the single-subject level and entered into SPM’s multiple regression analysis, in order to explore the effect of number of post-extinction CS+ offset-related vmPFC reactivations on day 2 on differential (CS+>CS-) changes in functional vmPFC–amygdala connectivity (all ROIs: Harvard-Oxford atlas; thresholded at 50% tissue probability) at test on day 3. We further included drug (placebo/L-DOPA) and its interaction with number of vmPFC reactivations into the second-level multiple regression model. Functional connectivity between the vmPFC and the amygdala during test on day 3 did not change in dependence of the number of post-extinction vmPFC reactivations after extinction on day 2 ($P > .05$; SVC, FWE). In order to preserve the clarity and (relative) simplicity of the manuscript, we would prefer to not include this additional analysis into the original manuscript. Finally, we would make this decision dependent on the reviewer’s accordence.

Second, do the vmPFC reactivations coincide with enhanced vmPFC-amygdala connectivity. The latter would fit with ideas about neural synchronization and memory replay from rodent electrophysiology research

Reply: This is indeed another interesting follow-up question. Practically, it is challenging to examine reactivation-specific changes in functional vmPFC connectivity, as the actual number of CS+ offset-related vmPFC reactivations differs greatly between participants (see Supplementary

Figure 2a), with some participants not showing any or only few reactivations. As the number of reactivations differed widely, the estimation of functional connectivity would be based on 0, 1, 2, 3, ... reactivations in some, and 20+ reactivations in other participants, making the functional connectivity beta-values incomparable between participants. However, it is straightforward to determine the functional vmPFC connectivity across the whole time course of a resting-state scan using standard seed-based resting-state analysis (seed region: vmPFC ROI, as defined for all other analyses) and to test whether a high number of potential vmPFC reactivations is associated with overall increased vmPFC-amygdala connectivity in the same resting-state scan. We, thus, conducted seed-based resting-state analysis with a GLM including each resting-state scan's mean vmPFC time-course, mean CSF and WM time course and the six motion regressors. We computed first-level contrasts testing the change in functional vmPFC connectivity from the pre-extinction resting-state scan to the 45 min post-extinction resting-state scan. Subsequently, we employed number of CS+ offset-related vmPFC reactivations 45 minutes after extinction as a regressor on the resulting first-level contrast image (vmPFC connectivity post 45 min > pre). The number of post-extinction vmPFC reactivations was, however, not related to vmPFC-amygdala connectivity in such an analysis ($P > .05$, SVC, FWE). The results do not change when conducting the same analysis on the post 45 min vmPFC functional connectivity beta-map, without subtracting the pre-extinction vmPFC functional connectivity.

- 2. The authors mention: "To correct for inter-individual variance, data were logtransformed (+1) and range-corrected within each subject and day". The potential problem is that this method might inflate small differences on day 3 and exaggerate differences between drug and placebo groups. If the authors wish to reduce interindividual variance then it would be advisable to apply log transformation and rescaling to the entire data set for a given participant, not independently for the different days.*

Reply: Thank you very much for raising this point. As a standard procedure in our laboratory we conduct log-transformation and range-correction within each subject and day in multiple-day paradigms in order to correct for changes between days in temperature, humidity, electrode placement, skin condition etc., that affect SCRs differentially within an individual across days. Note that, applying transformations equally to both placebo and drug group and to both CS+ and the CS- within each subject and day ensures the reliability of any potential drug effects on differential responding at test on day 3. However, we also followed the reviewer's suggestion and re-analyzed the SCR data using log-transformation and range-correction within each subject across days. Please find the according results based on data transformations across days in the figures below (Response Letter Figure 1). Critically, applying transformations across days does not change the main results and merely increases the effect size of the effect of drug on differential SCRs on day 3 (original partial $\eta^2 = .17$ vs. new partial $\eta^2 = .19$; Response Letter Figure 1a) and the effect size of the relation between post-extinction vmPFC reactivations and SCRs on day 3 ($\beta = -.13$ vs. $\beta = -.14$; Response Letter Figure 1b) slightly. The mediation effect remains similar ($\beta = -.09$ vs. $\beta = -.09$; Response Letter Figure 1c). As applying transformations across days does, however, not adequately correct for potential changes between days within an individual and as we would like to stay consistent with our previous work, we argue for keeping the

standard procedure of correcting SCRs within each day and hope that the reviewer agrees with the argumentation behind our decision.

Response Letter Figure 1 | Applying SCR log-transformation and range-correction across days does not change the main experimental effects or the effect of L-DOPA on CRs at test on day 3. (a)

On day 1, CRs were successfully acquired, as indicated by greater skin conductance responses (SCRs) to the CS+ than to the CS- in both groups (repeated measures ANOVA: stimulus: $F_{1,35}=61.54$, $P<.001$, partial $\eta^2=.64$, stimulus x group: $P=.96$; $n=37$ participants with sufficient SCR data quality, see Methods). On day 2, CRs were successfully retrieved at the beginning of extinction in both groups (stimulus: $F_{1,34}=24.86$, $P<.001$, partial $\eta^2=.42$; stimulus x group: $P=.79$; $n=36$). At the end of extinction learning, there was still a significant CS+>CS- difference on SCRs that did, however, not differ between groups (stimulus: $F_{1,34}=15.80$, $P<.001$, partial $\eta^2=.32$; stimulus x group: $P=.50$; $n=36$). As predicted, the post-extinction administration of L-DOPA compared to placebo (see Fig. 1a) enhanced extinction memory retrieval and reduced the expression of fear at test on day 3 (stimulus: $F_{1,33}=40.69$, $P<.001$, partial $\eta^2=.55$; stimulus x drug: $F_{1,33}=7.47$, $P=.01$, partial $\eta^2=.19$; $n=35$; Fig. 1b). Specifically, L-DOPA-treated participants showed significantly smaller SCRs to the formerly reinforced CS+ than placebo-treated participants (two-sample t test: $T_{33}=-2.53$, $P=.02$, Cohen's $d=.89$; $n=35$; Fig. 1b). Note, that there was only a significant difference between groups on mean SCRs across the test phase, but the speed of re-extinction did not differ significantly between drug groups (control analysis with stimulus (CS+, CS-) and trial (1-10) as within-, and drug (placebo, L-DOPA) as between-subject factor: stimulus x drug: $F_{1,297}=8.35$, $P=.007$, partial $\eta^2=.20$; stimulus x trial x drug: $F_{1,297}=1.84$, $P=.10$, Greenhouse-Geisser corrected; $n=35$). (b) The number of potential vmPFC pattern reactivations during the 45-min post-extinction scan negatively predicted CRs at test on day 3 (linear regression: $\beta_{45\text{min}}=-.11$, $SE=.02$, $T_{30}=-4.86$, $P=.00003$; $n=35$). The administration of L-DOPA did not affect the predictive relationship between number of CS+ offset-related vmPFC reactivations and CRs at test, but comparable relations between number of vmPFC reactivations and CRs at test were found in each group (multiple linear regression: interaction reactivations x drug, $P=.75$; simple slope placebo: $\beta=-.10$, $P=.005$, L-DOPA: $\beta=-.08$, $P=.04$; $n=35$). (c) Number of reactivations of the CS+ offset-related vmPFC pattern at 45 min post-extinction mediated the effect of L-DOPA on CRs at test

(mediation analysis: $\beta=-.09$, 95% CI: $-.16$ to $-.03$, $P=.004$; bootstrapping procedure with 10,000 simulations; $n=35$; Fig. 2g).

3. *The authors write that they: “ ... operationalized initial fear acquisition as SCRs to CSs averaged across the last 20% of trials on day 1, fear memory recall as SCRs to CSs averaged across the first 20% of trials on day 2 and fear at the end of extinction as SCRs to CSs averaged across the last 20% of trials on day 2. As in previous studies, the effect of L-DOPA on extinction memory retrieval was tested on SCRs to CS+ and CS-, averaged across the whole test phase on day 3”. It is not immediately clear from this description across how many trials averaging occurred (2 trials for some phases?). Could the authors also explain why the averaging for conditioning, extinction and recovery phases differed and whether there was an a priori rationale for this?*

Reply: At the outset of a series of three-day studies on the effects of a consolidation-enhancing effect of L-DOPA on fear extinction memory currently ongoing in our laboratory – including the present study – we defined the described analysis strategy as an a-priori analysis strategy for all studies, independent of the exact number of trials employed in an individual study during fear conditioning, extinction and test, respectively. The intention was to achieve harmonization and comparability across studies as well as to avoid a-posteriori hypothesis formulation. Because the number of conditioning, extinction and test trials may vary across studies depending on the experimental question, we defined the number of trials to be included in a score in terms of percentages. The exact percentage defined for days 1 and 2 is arbitrary, but the relatively small percentage (20%) expresses the dynamic nature of learning during conditioning and extinction. As for the 100% of trials chosen for the extinction memory test on day 3, we took into account experience from earlier studies (as mentioned; e.g. Haaker et al., 2013) and also wanted to exclude potential ephemeral effects of a manipulation that would only last for the initial stimulus presentations. The numbers of trials analyzed in the present experiment were the 2 last trials at the end of conditioning, the first and the last 3 trials during extinction and all 10 trials at test. However, we fully agree with the reviewer that this was not made sufficiently transparent to the reader in the previous version of the manuscript. We have now added the exact number of trials used for the analysis to the percentage on p. 22, l. 19 in the revised version of the manuscript.

Page 22, line 19: We operationalized initial fear acquisition as SCRs to CSs averaged across the last 20% of trials on day 1 (i.e. last 2 trials), fear memory recall as SCRs to CSs averaged across the first 20% of trials on day 2 (i.e. first 3 trials) and fear at the end of extinction as SCRs to CSs averaged across the last 20% of trials (i.e. last 3 trials) on day 2. As in previous studies^{17,19}, the effect of L-DOPA on extinction memory retrieval was tested on SCRs to CS+ and CS-, averaged across the whole test phase on day 3 (i.e. all 10 trials).

4. *The authors report that there was a difference in responses to the CS+ vs CS- at the beginning of extinction but also at the end of extinction. Did a Group (drug, placebo) x Phase (beginning, end of extinction) x CStype (CS+, CS-) ANOVA reveal a Phase x CStype interaction? This would provide evidence that extinction learning did, in fact, take place.*

Reply: Following the reviewer's suggestion we conducted a repeated measures ANOVA with group (placebo/L-DOPA) as between-, and phase (early/late) and stimulus (CS+/CS-) as within-subject factors. The analysis did not reveal any significant phase by stimulus interaction ($F_{1,34}=.001$, $P=.97$; $n=36$). Instead, only the main effect of phase was significant ($F_{1,34}=.49.45$, $P<.001$). That is, both CS+ and CS- evoked SCRs habituated across the extinction session, but differential SCRs were not fully reduced at the end of extinction, as indicated by the significant difference between CS+ and CS- evoked SCRs at the end of extinction (page 5, line 20). (Note, however, that this effect differed between individuals, with some showing full or very strong extinction and others not.) The lack of complete reduction of CRs at the end of extinction is particularly interesting in the light of the effect of L-DOPA on the expression of CRs at test on day 3. As mentioned in the introduction, "no or incomplete extinction may still protect against the return of CRs", that is, the longer-term success of extinction (or extinction memory retrieval) is not determined by the success of initial within-session extinction. This further substantiates our point of long-term extinction depending strongly on an independent consolidation process.

5. *The authors report that they look at neural activity at the offset of the conditioned stimulus. Although it is true that they set up separate regressors for the onset and offset of the CSs the duration of the CS is always 4.5 second and the regressors are thus temporally correlated. Based on this one cannot conclude that the modeled activity purely reflects offset-related responses.*

Reply: We thank the reviewer for raising this important issue. Previous work has strongly emphasized the importance of controlling multi-collinearity between regressors when conducting representational similarity analysis in event-related designs (Mumford *et al*, 2012, 2014; Visser *et al*, 2016). In the employed GLM we included one CS onset regressor comprising all 15 CS onsets across the entire extinction session, one CS offset regressor comprising the first 5, and one CS offset regressor comprising the last 10 CS offsets (for both CS+ and CS-, respectively). Thereby, the two CS offset-related regressors are indeed temporally suspended to the CS onset regressor by 4.5 sec, however, one relates to the first 5 and one to the last 10 CSs, respectively. Thus, the average Pearson correlation coefficients between all CS onset and all CS offset regressors convolved with the hemodynamic response function are small (range: $R=.003-.056$). In order to provide the reader with this information we have now included the post-convolution regressor correlations in Supplementary Figure 9 and refer to it in the description of the GLMs in the Methods section on p. 24, l. 4.

Supplementary Figure 9 | Pearson correlation coefficients between CS on- and offset related regressors in the general linear models (GLM) used for the main and the control analysis. The short interval between CS on- and offset of only 4.5 sec raises the possibility that the CS+ offset-related regressor used for the reactivation analysis may be affected by high collinearity with the CS+ onset-related regressor. **a)** However, in the GLM used for the main analysis the Pearson correlation coefficients between all regressors convolved with the hemodynamic response function (HRF) were low on average. Importantly, the regressor corresponding to the first 5 CS+ offsets was not significantly correlated with the regressor corresponding to the CS+ onsets ($R=.04$, $P>.05$; $n=40$). **b)** Similarly, in the GLM used for the control analysis the regressors corresponding to the first, middle or last 5 CS+ offsets were not correlated with the regressor corresponding to the CS+ onset (all $R_s<.04$, $P_s>.05$; $n=40$).

Page 24, line 4: Even though CS on- and CS offset-related regressors succeeded other by 4.5 seconds only, correlations between the HRF-convolved regressors were low (Supplementary Figure 9).

What's more, this raises the question if the authors would use a neural signature of the CS onset they would find similar results? If so, this would also speak to the next point.

Reply: This is an interesting question and we agree with the reviewer that this possibility should be excluded. Following the reviewer's suggestion to test whether reactivations of a CS+ **onset**-related vmPFC activity pattern would yield similar results, we first employed the CS+ onset-related vmPFC activity pattern from the original GLM used in the main text (comprising: CS+ onset regressor, CS- onset regressor, first 5 CS+ offset regressor, last 10 CS+ offset regressor, first 5 CS- offset regressor, last 10 CS- offset regressor, context on-/offset, ratings). There was no relation between CS+ onset-related vmPFC reactivations and CR (SCR CS+>CS-) at test at any post-extinction resting-state scan (linear regression - CS+ onset: $\beta_{10min}=.00$, $P=.87$, $\beta_{45min}=-.00$, $P=.92$, $\beta_{90min}=.01$, $P=.60$). However, it is conceivable that in this GLM the inclusion of the CS offset-related regressors may have picked up some CS related-variance. Furthermore, it is conceivable that CS onset-related vmPFC reactivations from a specific phase during the extinction session may be relevant for extinction consolidation (e.g. fear memory retrieval related CS onsets early in extinction or CS onsets late in extinction presumably related to the acquisition of the CS-noUS association). Thus, we also re-analyzed the data employing a GLM with one regressor for the first 5 CS+ and CS- onsets each, one for the middle 5 CS+ and CS- onsets each, one for the last 5 CS+ and CS- onsets each, ratings and context on-/offsets. We did **not** include any CS **offset** regressors into this GLM, in order to allow the CS onset regressors to pick

up all CS-related variance. There was no relation between early, middle or late CS+ onset-related vmPFC activity pattern reactivations at any post-extinction resting-state scan and CRs at test on day 3 (linear regression - first 5 CS+ onset: $\beta_{10\text{min}}=-.06$, $P=.07$, $\beta_{45\text{min}}=-.03$, $P=.43$, $\beta_{90\text{min}}=-.05$, $P=.22$; linear regression - middle 5 CS+ onset: $\beta_{10\text{min}}=-.05$, $P=.13$, $\beta_{45\text{min}}=-.03$, $P=.42$, $\beta_{90\text{min}}=-.03$, $P=.28$; linear regression - last 5 CS+ onset: $\beta_{10\text{min}}=-.06$, $P=.12$, $\beta_{45\text{min}}=-.03$, $P=.51$, $\beta_{90\text{min}}=-.02$, $P=.48$; all $n=35$). Based on these results, we are confident that the reported relation between vmPFC reactivations and extinction memory retrieval 24 h later is specific to vmPFC activity patterns evoked at CS+ offset early in extinction. We now include these results in Supplementary Figure 5 (see below) and refer to them in the main text.

Supplementary Figure 5 | The effects are specific to reactivations of the CS+ offset-related vmPFC activity pattern early in the extinction session. (a) The number of potential spontaneous reactivations (45 min post-extinction) of the CS- offset related vmPFC pattern (i.e. first 5 CS- offsets for which US omission is expected by the participant) does not predict CR at test (linear regression: $\beta_{45\text{min}}=-.02$, $\text{SE}=.04$, $T_{33}=-.30$, $P=.76$; $n=35$). There was also no relation between spontaneous CS- offset-related vmPFC reactivations at 10 or 90 minutes after extinction learning and CR at test (data not shown, $P_s>.14$). Similarly, there was no relation between the number of potential spontaneous reactivations (45 min post-extinction) of the pattern elicited by the CS+ offsets during **(b)** the middle (i.e. 6th-10th trial; linear regression: $\beta_{45\text{min}}=-.02$, $P=.63$; $n=35$) and **(c)** the end of the extinction session (11th-15th trial; linear regression: $\beta_{45\text{min}}=-.05$, $P=.15$; $n=35$) (i.e. where repeated US omission has been experienced by the participant) and CR at test. Spontaneous CS+ offset-related vmPFC reactivations from the middle or end of extinction learning reactivation at 10 or 90 minutes after extinction learning were also not related to CR

at test (data not shown, $P_s > .15$). **(d)** Critically, the effects are also not observed for reactivations (45 min post-extinction) of the first 5 CS+ onset-related vmPFC activity patterns (linear regression: $\beta_{45\text{min}} = -.03$, $SE = .03$, $T_{33} = -.79$, $P = .43$; $n = 35$). There was also no relation between spontaneous CS- offset-related vmPFC reactivations at 10 or 90 minutes after extinction learning and CR at test (data not shown, $P_s > .07$). **(e)** Similarly, the effects are also not observed for reactivations (45 min post-extinction) of the middle 5 CS+ onset-related vmPFC activity patterns (linear regression: $\beta_{45\text{min}} = -.03$, $SE = .04$, $T_{33} = -.81$, $P = .42$; $n = 35$). There was no relation between spontaneous middle 5 CS+ onset-related vmPFC reactivations at 10 or 90 minutes after extinction learning and CR at test (data not shown, $P_s > .13$). **(f)** Finally, the effects are also not observed for reactivations (45 min post-extinction) of the middle 5 CS+ onset-related vmPFC activity pattern (linear regression: $\beta_{45\text{min}} = -.03$, $SE = .04$, $T_{33} = -.66$, $P = .51$; $n = 35$). There was no relation between spontaneous middle 5 CS+ onset-related vmPFC reactivations at 10 or 90 minutes after extinction learning and CR at test (data not shown, $P_s > .12$).

Page 7: Importantly, the relation was also specific to the pattern elicited by the unexpected US omission at CS+ offset, as opposed to the expected US omission at CS- offset (Supplementary Fig. 5a), to the US omissions at CS+ offset during later phases of extinction (middle and last 5 trials; Supplementary Fig. 5b and c), at which the omission of the US was not surprising to the participant anymore, or to the CS+ onset in early, middle or late extinction (Supplementary Fig. 5d-f).

- 6. The authors state that the extracted vmPFC data reflects prediction error related activity because “Early in the extinction session, US omission is unexpected to the participants and elicits a prediction error.” Although this is true, any trial in a conditioning experiment will evoke a prediction error, albeit of a different amplitude. Throughout the literature the use of the term “prediction error” is consistently being inflated. The term “prediction error” stems from formal reinforcement learning models and reflects a specific parametric value that is not estimated in the current submission. The signal extracted by the authors could equally reflect “novelty”, “surprise”, “entropy”, “associability”, simple “arousal” or anything else. Interestingly, the authors discuss the work by Peyrache but seem to have missed the follow up work by Benchanane which shows that dopamine in the mPFC is associated with increases “memory replay” but in these tasks this is clearly independent of prediction errors. Hence, the link between dopamine, prediction error, and memory reactivations is far from clear. My suggestion is to omit the use of this term and just mention that the focus is on neural activity at the offset of the stimulus as this is the time-point at which learning is likely to occur. Leave discussion of the possible link to prediction errors to the discussion.*

Reply: The reviewer is correct. We have now changed the phrasing in the main text of the revised manuscript and replaced ‘prediction error-related vmPFC activity pattern’ with ‘CS offset-related vmPFC activity pattern’ in the presentation of the results. We now only raise the possibility of an involvement of a prediction error signal in establishing the to-be-reactivated vmPFC activity pattern in the discussion.

Page 11, line 3: Even though not modeled explicitly in the present study, the unexpected omission of the aversive US early in extinction may elicit a prediction error signal (“outcome better than expected”) initiating new learning.

7. *The authors report that L-dopa increased the number of detected reactivation 45min but not 10 min or 90 min following extinction training. They also mention that peak plasma concentrations of L-dopa occur after 45 min, which may explain why L-dopa enhanced reactivation are not found after 10 minutes it does not explain why L-dopa does not enhance reactivations after 90 minutes. If there truly is a reason that reactivation occur after 45 minutes but not 10 or 90 minutes then there should be an interaction effect of Drug x Rest scan. Such a result would strengthen the finding.*

Reply: We apologize for the unclear presentation of results. In a repeated measures ANOVA with resting-state scan as within and group as between subject factor, there is indeed a significant Drug x Rest Scan interaction (see page 8 of the original manuscript: time x drug: $F_{3,114}=3.51$, $P=.02$, partial $\eta^2=.09$). Post-hoc *T*-Tests reveal that L-DOPA significantly increased number of reactivations at 45 min after extinction ($T_{38}=-3.05$, $P=.004$, Cohen's $d=.97$), but there was also trend-wise significant effect 90 minutes after extinction ($T_{38}=-1.95$, $P=.06$, Cohen's $d=.62$). The results of the repeated measures ANOVA were given on page 8 of the original manuscript and the result of the post-hoc *T*-tests in the legend of Figure 2. Thanks to the comment of the reviewer, we became aware that this presentation of the results was not optimal and we do present them now all on page 8 of the revised manuscript (see below).

Page 8: Indeed, post-extinction L-DOPA administration significantly increased the number of potential CS+ offset-related vmPFC pattern reactivations (time x drug: $F_{3,114}=3.51$, $P=.02$, partial $\eta^2=.09$; $n=40$), particularly 45 minutes after extinction and with a trend-wise significant effect 90 minutes after extinction (post-hoc two-sample *t* test; pre: $P=.39$; post: 10 min: $P=.09$; 45 min: $T_{38}=-3.05$, $P=.004$, Cohen's $d=.97$; 90 min: $T_{38}=-1.95$, $P=.06$, Cohen's $d=.62$; $n=40$; Fig. 2e).

8. *In the mediation analyses it is unclear to me exactly which relationships are being described. The beta estimate mentioned in the main text does not reappear in Figure 2g. Mediation analyses generally examine and report the direct relationship between the predictor (drug) and criterion (SCR), the relationship between the predictor the mediator (reactivations), the relationship between the mediator and the criterion, and critically whether the direct relationship between the predictor and criterion is reduced when controlled for the relationship between the mediator and criterion. Could the authors more comprehensively describe their methods and results?*

Reply: We apologize for the non-transparent presentation of the results. The critical paths between predictor, mediator and criterion, referred to by the reviewer, were provided in Figure 2g of the original manuscript (for individual paths, please see adapted Figure 2g below):

Path **a**: $\beta=.97$, the direct relationship between predictor (drug) and mediator (reactivations).

Path **b**: $\beta=-.10$, the direct relationship between mediator (reactivations) and criterion (SCR).

Path **c**: $\beta=-.15$, the direct relationship between the predictor (drug) and the criterion (SCR).

Path **c'**: $\beta=-.06$, the direct relationship between the predictor (drug) and the criterion (SCR) after inclusion of the mediator (reactivations) into the model.

The **beta-estimate in the main text** (i.e. $\beta = -.09$, 95% CI: $-.13$ to $-.02$, $P = .007$), referred to by the reviewer, represents the actual mediation effect, i.e. the indirect effect between the predictor (drug) and the criterion (SCR) via the mediator (reactivations). It is revealed by the reduction of path c from before (path **c**: $\beta = -.15$) to after (path **c'**: $\beta = -.06$) the inclusion of the mediator into the model (**c-c'** = $-.09$). In order to make the applied analysis transparent to the reader, we have now labeled the effect referred to in the main text as “indirect, mediator effect”, labeled the individual paths in Figure 2g and included the detailed results for each path in the legend of Figure 2g (see below). We also noted that the number of stars in Figure 2g indicating p-value size was not accurate. We apologize for this mistake and have corrected it now. We hope we could thereby clarify the applied analysis and the presentation of the mediation results.

g

Figure 2 | Spontaneous post-extinction reactivations support extinction memory consolidation.

[...] **g**) The post-extinction administration of L-DOPA had a significant positive effect on number of potential CS+ offset-related vmPFC reactivations (path a: $\beta = .97$, $SE = .30$, $T_{33} = 3.23$, $P = .003$; $n = 35$). The number of potential CS+ offset-related vmPFC reactivations 45 min after extinction was significantly negatively related to smaller differential CRs at test on day 3 (path b: $\beta = -.10$, $SE = .03$, $T_{33} = -4.26$, $P = .0002$; $n = 35$). After inclusion of number of potential CS+ offset-related vmPFC reactivations into the latter model, the significant effect of drug on CRs at test (path c: $\beta = -.15$, $SE = .06$, $T_{33} = -2.57$, $P = .02$; $n = 35$) decreased (path c': $\beta = -.06$, $SE = .06$, $T_{32} = -.98$, $P = .33$; $n = 35$), indicating that the effect of L-DOPA on CRs at test on day 3 was significantly mediated (**c-c'** = $-.09$, 95% CI: $-.13$ to $-.02$, $P = .007$; bootstrapping procedure with 10.000 simulations; $n = 35$) by number of potential CS+ offset-related vmPFC reactivations 45 min after extinction.

Page 8: Intriguingly, the number of reactivations of the CS+ offset-related vmPFC pattern at 45 min post-extinction mediated the effect of L-DOPA on CRs at test (Fig. 2g; mediation analysis – indirect mediator effect: $\beta = -.09$, 95% CI: $-.13$ to $-.02$, $P = .007$; bootstrapping procedure with 10.000 simulations; $n = 35$).

In addition, the authors exaggeration as the beta-value is low and unlikely to explain all variance in SCR. They use similar terminology (“complete”) in the discussion and I would also suggest removing it there.

Reply: We thank the reviewer for this helpful remark. We were not aware of the fact that our usage of the terminology “full” or “complete” mediation was based on an outdated classification of mediation effects (Baron and Kenny, 1986) stating that a non-significant beta-estimate of the

c' path after inclusion of the mediator into the model sufficiently indicates "full" or "complete" mediation. Based on the valuable remark of the reviewer, we have now changed the terminology across the whole manuscript and omit the terms "completely mediated" or "fully mediated" entirely.

9. *It would be useful if the authors would report effect sizes. The journal might even require this.*

Reply: We now report effect sizes for the results of repeated measures ANOVA (partial η^2) and *T*-Tests (Cohen's *d*) across the manuscript and supplementary information. In multiple linear regression analysis, unstandardized or standardized beta-coefficients provide an estimate of the size of the effect. The advantage of unstandardized beta-coefficients is that they can be interpreted in respect to the actual unit of the outcome measure. That is, an increase of 1 (log-transformed) vmPFC reactivation is associated with a decrease of $\beta_{45\text{min}} = -.13$ in differential SCR (CS+>CS-) during test on day 3 in our main regression model (cf. manuscript, page 6). Alternatively, when using the non log-transformed number of vmPFC reactivations (cf. Supplementary Figure 2) the effect is even more clear: an increase of 1 post-extinction vmPFC reactivation is associated with a decrease of $\beta_{45\text{min}} = -.02$ in differential SCR (CS+>CS-) during test on day 3. As CS+ and CS- evoked SCRs can range between 0 and 1 only (due to the applied range-correction), we think that the unstandardized beta-coefficients provide a relatively intuitively interpretable measure of effect size in the present regression analyses. We have therefore settled on providing the individual unstandardized beta-coefficients as a measure of the effect of the individual coefficients in regression analysis and hope the reviewer agrees with this decision.

Suggestions:

1. *Be careful with the clinical framing. First, because there are effective evidence-based treatments for stress and anxiety disorders and these are not all or completely based on extinction learning. Second, because the translation of fundamental research to clinical application for the treatment of psychiatric disorders has proven difficult. Related, there is ongoing debate whether conditioned defensive responses truly reflect subjective fear. This debate is especially relevant in the translation from fundamental research to clinical application. Considering the authors clinical framing of the manuscript and use of the terminology "fear-conditioning" I would advise the authors to acknowledge this.*

Reply: We have carefully re-examined the parts of the text where we referred to clinical observations. The formulation on p. 3, l.1 and p.10, l. 9 of the original manuscript ("Fear extinction is believed to protect against the development of stress-related pathology after trauma¹."; "Fear extinction promotes resilience against the development of post-traumatic stress orders^{1a}") is based on predictions of mental health outcomes after trauma based on actual laboratory extinction tests. The formulation on p. 3, l. 2 "Furthermore, exposure interventions based on the principles of extinction learning are a cornerstone of cognitive-behavioral therapy of anxiety disorders and post-traumatic stress disorder (PTSD)²." correctly reflects the historical development of exposure interventions. We therefore think these sentences can be maintained. However, in the original formulation on p.3, l. 12: "An important problem that plagues extinction

research and therapeutic practice is that the success of extinction learning is neither a strong nor reliable predictor of the long-term retrieval of the extinction memory^{4,5}.” we indeed generalized from extinction learning in the laboratory to “therapeutic practice”, where extinction is only thought (but not proven) to underlie the effects of exposure. We therefore now omit “and therapeutic practice” in the new manuscript version. The formulation on p. 10, l. 9 “Fear extinction ... is employed during cognitive-behavioral therapy in order to reduce pathological fears⁵.” was changed to “ and fear extinction principles are employed during cognitive-behavioral therapy in order to reduce pathological fears⁵ .”

2. *The authors write that they US is a “painful stimulus”, shouldn’t this simply be aversive or annoying? The intensity at which electrical stimulation becomes painful is quite high and in my experience participants never allow the researcher to set the stimulator that high, nor would one want to as it might be dangerous and unethical.*

Reply: Indeed, US calibration procedures in many other laboratories involve calibration to a level perceived as “uncomfortable, but not yet painful”. In contrast, in our laboratory USs are calibrated to a level that is perceived as “maximum tolerable pain” by the participant (e.g. Haaker *et al*, 2013; Lonsdorf *et al*, 2014). That is, on a scale from 0 (= “I do not feel anything”) to 10 (= “strongest pain imagined to be deliverable via such an electrode”), we aim to calibrate the US to a subjective strength of 6-7 in all participants (see Supplementary Table 1). We believe that employing actual painful USs increases the ecological validity of aversive conditioning paradigms in humans and allows for a more direct translation of results between human and animal studies, in which electric shocks delivered via a grid to the animal’s paws are truly aversive stimuli. While it is true that we cannot exclude with certainty that the final level chosen by a participant is not actually painful to him or her, pain is an inherently subjective construct and we can therefore only rely on the subject’s self-report (as in any study employing pain induction).

3. *The experiment uses an ABB paradigm. Although it is questionable if background stimuli truly trigger contextual processing in humans and aren’t simply compound stimuli, it might be interesting for the authors to discuss to what extent DA is effecting the cue based extinction memory trace or the context or compound memory? For example, we have recently shown that a noradrenergic manipulation during extinction learning using a similar ABB paradigm may effect contextual safety learning (Kroes *et al.*, 2015 *Neuropsychopharmacology*).*

Reply: This is indeed an interesting question. Given the present paradigm it is difficult, though, to discern whether L-DOPA affected the cue-based extinction memory or the context/compound memory, as we presented stimuli on day 3 in context B only and did not include any additional tests on day 3 in which the context/compound stimulus was manipulated independent of the cue stimulus. It is conceivable, though, that L-DOPA may have led to a reduction of SCRs to the presentation of the extinction context alone, due to a potential enhancement of the contextual extinction memory trace. Thus, we analyzed the SCR evoked by the context onset before the presentation of any stimulus during test on day 3. There was no significant difference in the context onset-evoked SCRs between placebo and L-DOPA treated participants (two-sample *T*-test: $T_{33}=-.96$, $P=.34$; $n=35$), indicating that the effect of L-DOPA may have rather been limited to

the presentation of the cue in the extinction context. Please note, however, that the context onset-related SCR refers to one single SCR for each participant, which is strongly affected by orienting responses due to the sudden switch between the presentation of a blank screen to the presentation of the context picture. Given that the paradigm was not optimized to answer this question, we hope the reviewer agrees with our decision to not include a discussion on L-DOPA's potential differential effects on cue vs. context/compound extinction memory into the manuscript.

4. *The authors write that: "Our results extend these findings by providing first evidence for a causal role of fMRI pattern reactivations in human memory consolidation". Although it is true that the authors show that L-dopa administration is associated with neural reactivation patterns, establishing causality in neuroscience -and especially with fMRI- is though. I suggest refraining from using this term as it is not necessary to describe the observed phenomena.*

Reply: We agree with the reviewer that fMRI research does not allow for establishing causality. We have changed the phrasing of the respective (p. 12, l. 2) and other sentences referring to causal evidence based on fMRI results as follows.

Page 5, line 7: These results provide first evidence for a critical role of dopamine-dependent vmPFC activity pattern reactivations in the consolidation of human extinction memories

Page 10, line 3: This is the first evidence in the human memory literature for a dopamine-dependent amplification of neural pattern reactivations during memory consolidation, as detected in fMRI.

Page 12, line 2: Our results extend these findings by providing first evidence for a critical role of dopamine in fMRI pattern reactivations during human memory consolidation.

Page 13, line 14: By experimentally manipulating dopaminergic activity after learning and observing increases in spontaneous vmPFC reactivation frequency associated with improved extinction memory retrieval, the present study further provides first evidence for a critical role of dopamine in human memory reactivations.

Regardless of these comments, I am fascinated by the results and enjoyed reading the manuscript.

*Sincerely,
Marijn Kroes
(I sign all my reviews)*

Reviewer #3 (Remarks to the Author):

In the reported fMRI pavlovian fear conditioning study, the authors address two questions: (1) whether, within VMPFC, cross-voxel activation patterns associated with CS presentation and US omission during early extinction can be observed during later resting state scans (in line with consolidation of these representations) and whether the number of such 'reactivations' inversely predicts the strength of conditioned responses at test (extinction memory recall) the day after, and (2) whether administration of L-DOPA after extinction training increases the number of CS-no US reactivations in VMPFC and if this mediates the effect of L-DOPA on extinction memory (i.e. reduced conditioned responses) at test.

The authors give a comprehensive and valuable, if somewhat dense, introduction. They detail how they build on the rodent literature where number of infralimbic bursts during consolidation of extinction training predicts extinction memory retrieval at test 24 hours later. To translate this into a paradigm suitable for fMRI with human participants, the authors take advantage of methodology developed by Staresina et al. 2013. The authors of this prior paper showed how multivoxel pattern analysis could be used to compare activation patterns at encoding with activation patterns during each volume of a later resting state scan. They further showed that reactivation of encoding patterns was stronger for items later recalled than for items later forgotten. The current authors have nicely drawn on this clever paradigm to address the question of whether the representations encoded during extinction training (namely the pairing of the CS+ with the non-occurrence of the US) are similarly 'replayed' during rest and if the extent of such replay can predict extinction memory and explain the influence of LDOPA upon extinction memory. The study is hence both theoretically well motivated and methodologically well designed (though see comment 1 on modeling below). The results are compelling and the discussion well written. I believe this paper will make a valuable contribution to the field.

Comments

Major

- 1) Regressors used in modeling the extinction learning data. The authors used the following regressors in their GLM: " one regressor for CS+ onsets, CS-onsets, pre- and post-extinction US expectancy ratings and context on-/offset each. In addition, the model included one regressor for the first 5 CS+ offsets, at which the omission of the US is unexpected, and the first 5 CS- offsets, at which US omission is expected by the participant. In addition, one regressor for the remaining 10 CS+ and one regressor for the remaining 10 CS- offsets were included. For control analyses (Supplementary Fig. 5B and C) we created an additional single-subject level model including CS+ onsets, CS-onsets, US expectancy ratings, context on-/offsets and one regressor each for the first, middle and last 5 CS+ and CS- offsets, respectively."*

I am concerned that given the relatively short fixed duration of the CS+ (4.5s) there is likely to be substantial colinearity between the CS+ onset regressor and the 2 CS+ offset regressors in the main model (after convolution with the hrf), similarly for the CS- onset regressor and the 2 CS- offset regressors. The same goes for the control analysis model.

Please can the authors give the post-convolution correlations between regressors in a supplementary figure. If there is indeed high correlation (and hence shared variance) between the CS onset and offset regressors, the authors need to address how this impacts their analysis.

Reply: We want to thank the reviewer for her/his supportive comments on our study and for raising this important question. Previous work has emphasized the importance of controlling multi-collinearity between regressors when conducting representational similarity analysis in event-related designs (e.g. Mumford *et al*, 2012, 2014; Visser *et al*, 2016). In order to assess the strength of collinearity between the CS onset and offset regressors, we followed the reviewer's suggestion and computed correlation coefficients between the CS onset and the CS offset regressors (after convolution with the HRF) for both the main and the control GLM referred to by the reviewer. Pearson correlation coefficients between convolved regressors were generally low (range main model: .003-.056; range control model: .002-.055; see Supplementary Figure 9 below). Specifically, in the main GLM (Supplementary Figure 9a), correlation coefficients between the CS+ onset regressor and the first 5 CS+ offset regressors or the last 10 CS+ offset regressors in the main GLM were $R=.035$ and $R=.049$, respectively. Similarly, in the control GLM (Supplementary Figure 9b), correlation coefficients between the CS+ onset regressor and the first 5, middle 5 or last 5 CS+ offset regressors were $R=.0362$, $R=.0363$ and $R=.0357$, respectively. In order to provide the reader with this information we have now included the post-convolution regressor correlations in Supplementary Figure 9 and refer to it in the description of the GLMs in the Methods section on p. 24, l. 8.

Supplementary Figure 9 | Pearson correlation coefficients between CS on- and offset related regressors in the general linear models (GLM) used for the main and the control analysis. The short interval between CS on- and offset of only 4.5 sec raises the possibility that the CS+ offset-related regressor used for the reactivation analysis may be affected by high collinearity with the CS+ onset-related regressor. **a)** However, in the GLM used for the main analysis the Pearson correlation coefficients between all regressors convolved with the hemodynamic response function (HRF) were low on average. Importantly, the regressor corresponding to the first 5 CS+ offsets was not correlated with the regressor corresponding to the CS+ onsets ($R=.04$, $P>.05$; $n=40$). **b)** Similarly, in the GLM used for the control analysis the regressors corresponding to the first, middle or last 5 CS+ offsets were not correlated with the regressor corresponding to the CS+ onset (all $R_s<.04$, $P_s>.05$; $n=40$).

Page 24, line 4: Even though CS on- and CS offset-related regressors succeeded other by 4.5 seconds only, correlations between the HRF-convolved regressors were low (Supplementary Figure 9).

- 2) *My second comment/concern also relates to the authors' estimation of the CS no US activation pattern at encoding. The authors chose to average activation at CS+ offset across the first five (of 15) trials where the CS+ was presented without the US. They justify this as follows: "Early in the extinction session, US omission is unexpected to the participants and elicits a prediction error. Prediction errors drive the correction of the US prediction associated with the CS and induce extinction learning. To capture this time point, we extracted the average multivoxel vmPFC activity pattern evoked by the first 5 offsets of the CS+ (the CS that had previously been paired with the US during conditioning) during extinction." The authors return to this in the discussion to argue that "it is conceivable that a prediction error signal elicited by the unexpected omission of the US in the mesostriatal dopamine system early in extinction learning induces the creation of a representation of the CS-noUS association or a representation of the latent 'extinction cause' in the vmPFC, which is then reactivated during extinction memory consolidation." Here the authors are arguing that the early trials are key because these are where the prediction error will be strongest and it is this error which induces the CS-no US representation which is later reactivated. However averaging across the first five trials is a fairly crude way to capture the extent of the prediction error on each trial. This would be easily modeled, allowing for the pattern on each CS+ no US trial to be weighted accordingly. This is unlikely to dramatically change the results but would better fit the current discussion/framing of the results as given above.*

Reply: We thank the reviewer for this important remark. We agree that estimating the vmPFC activity pattern based on the first 5 trials is a fairly crude approach (please also see our response to reviewer 1). In our hands individual SCRs are noisy and do not allow for a reliable estimation of prediction error time courses for each individual participant. We therefore decided to not employ the recorded trial-by-trial SCR for modelling. However, the reviewer raises an important point, and we believe future studies should be designed to include modeling. This could be achieved by employing an identical stimulus presentation order in all subjects, allowing for averaging SCR time courses and conducting modeling on these averaged time-courses. As for the current manuscript, we agree with the reviewer that modeling prediction error time courses and estimating the vmPFC activity pattern based on such time courses would be a prerequisite for the current discussion/framing. Therefore, we have now changed the phrasing in the main text of the revised manuscript and replaced 'prediction error-related vmPFC activity pattern' with 'CS offset-related vmPFC activity pattern' in the presentation of the results. We only refer to the possibility of an involvement of the prediction error in establishing the respective vmPFC activity pattern in the discussion. There we also acknowledge the limitation of our study to not include modeled prediction error time courses there (p. 11, l. 4). We hope that the reviewer agrees with this procedure.

Page 11, line 3: Even though not modeled explicitly in the present study, the unexpected omission of the aversive US early in extinction may elicit a prediction error signal ("outcome better than expected") initiating new learning.

3) *An initial concern of mine, while reading the paper, was that a few methodological choices seemed fairly arbitrary and I worried whether the results obtained would have been robust to different choices. For example, the authors use the first 5 trials of extinction to represent the early phase of extinction learning for the BOLD activation patterns (and the last 5 to represent late extinction training) but only the first / last 3 trials for the skin conductance data. It would be helpful if the authors could show what would have been obtained if the first (and last) 5 trials had been used for the skin conductance analyses too. If they feel it is important to only use 3 trials a justification for this should be given. Similarly, why did the authors use all trials at extinction memory test? I note that in other places where decisions might have seemed arbitrary, the authors have done a great job in the supplementary figures of showing how the decisions taken did not impact the results obtained. This went a long way to strengthening my faith in the results.*

Reply: At the outset of a series of three-day studies on the effects of a consolidation-enhancing effect of L-DOPA on fear extinction memory currently ongoing in our laboratory – including the present study – we defined the described analysis strategy as an a-priori analysis strategy for all studies independent of the exact number of trials employed in an individual study during fear conditioning, extinction and test, respectively. The intention was to achieve harmonization and comparability across studies as well as to avoid a-posteriori hypothesis formulation. Because the number of conditioning, extinction and test trials may vary across studies depending on the experimental question, we defined the number of trials to be included in a score in terms of percentages. The exact percentage defined for days 1 and 2 is arbitrary, but the relatively small percentage (20%) expresses the dynamic nature of learning during conditioning and extinction. As for the 100% of trials chosen for the extinction memory test on day 3, we took into account experience from earlier studies (as mentioned; e.g. Haaker et al., 2013) and also wanted to exclude potential ephemeral effects of a manipulation that would only last for the initial stimulus presentations. However, we agree that it is critical that the results do not depend on the exact number of trials employed. Thus, we repeated the SCR analysis using the first/last 5 trials during extinction on day 2. Importantly, these changes did not affect the results (see results reported below). In order to stay consistent with the analysis of the three-day L-DOPA studies in our laboratory and to simplify the presentation of the results, we do have a slight preference for not including these additional control analyses into the manuscript. Finally, we leave the decision to the reviewer.

Day 2: test of stimulus and group effect at the **beginning of extinction** (with first 3 trials: stimulus: $F_{1,34}=29.87$, $P<.001$, partial $\eta^2=.47$; stimulus x group: $P=.36$; with first 5 trials: stimulus: $F_{1,34}=49.65$, $P<.001$, partial $\eta^2=.59$; stimulus x group: $P=.55$; $n=36$).

Day 2: test of stimulus and group effect at the **end of extinction** (with first 3 trials: stimulus: $F_{1,34}=14.49$, $P<.001$, partial $\eta^2=.30$; stimulus x group: $P=.54$; with first 5 trials: stimulus: $F_{1,34}=23.65$, $P<.001$, partial $\eta^2=.41$; stimulus x group: $P=.78$; $n=36$).

4) *Figure 2c and Fig S2, S3 – please do not use uncorrected thresholds to show small volume activations – please use the corrected thresholds as in the main text. Otherwise the activations displayed are misleading.*

Reply: In previous publications, we differentiated between the threshold used for statistical inference (reported in the text) and the threshold used for display (in the figures), which serve two different purposes. A liberal and unmasked display threshold in our eyes primarily serves to show the wider distribution of activity, also beyond the ROI chosen for statistical testing. The advantage of showing a wider distribution at a relatively liberal threshold is that it allows for assessing whether activity in a ROI truly stems from that ROI or whether it rather stems from neighboring areas, “bleeding” into the ROI and thereby producing an artificial result. Please also note that the SVC/FWE threshold used for inference only refers to the chosen ROI, whereas it does not apply to activity in other regions also visible in a graph and can therefore not be used to meaningfully illustrate activation there. In figure 2c of the original manuscript, we had therefore chosen a more liberal unmasked display threshold of $P < 0.001$ that demonstrates the regional specificity of the vmPFC activation (see Response Letter Figure 2 below). For comparison, please also find an adjusted figure 2c below (Response Letter Figure 3) in which activity is thresholded at the statistical inference threshold of $P < 0.05$ SCV, FWE. Even though the employed display threshold does not affect the presentation of the results strongly, we would argue for employing a more liberal display threshold of $P < .001$ to provide the reader with more information about the extent of the activation. We do, however, also acknowledge the reviewer’s argument and would leave the final decision on the to-be-employed display threshold to the reviewer.

Response Letter Figure 2 | Relation between potential post-extinction CS+ offset-related vmPFC reactivations and vmPFC activity during test 24 h later, **thresholded at $P < .001$, uncorrected**, no masking applied.

Response Letter Figure 3 | Relation between potential post-extinction CS+ offset-related vmPFC reactivations and vmPFC activity during test 24 h later, **thresholded at $P < .05$, SVC, FWE**, no masking applied.

Minor

- 1) *Please refer to last 3 trials not last 20% of trials – otherwise the reader has to pause, go to the methods to find the number of trials and figure out how many 20% equates to – it puts unnecessary work on the reader.*

Reply: This phrasing is indeed not sufficiently transparent. We have now added the following information to the revised manuscript (p. 22, l. 19):

Page 22, line 19: We operationalized initial fear acquisition as SCRs to CSs averaged across the last 20% of trials on day 1 (i.e. last 2 trials), fear memory recall as SCRs to CSs averaged across the first 20% of trials on day 2 (i.e. first 3 trials) and fear at the end of extinction as SCRs to CSs averaged across the last 20% of trials (i.e. last 3 trials) on day 2. As in previous studies^{17,19}, the effect of L-DOPA on extinction memory retrieval was tested on SCRs to CS+ and CS-, averaged across the whole test phase on day 3 (i.e. all 10 trials).

- 2) *'Fully mediated' and 'complete mediation' implies the entire relationship can be explained by (i.e. 100% of variance); the authors should be careful in using this term as opposed to 'significantly mediated'*

Reply: We thank the reviewer for pointing out this issue. We were not aware of the fact that our usage of the terminology “full” or “complete” mediation was based on an outdated classification of mediation effects (Baron and Kenny, 1986) stating that a non-significant beta-estimate of the c' path after inclusion of the mediator into the model sufficiently indicated “full” or “complete” mediation. Based on the valuable remark of the reviewer, we have now changed the terminology across the whole manuscript and omit the terms “completely mediated” or “fully mediated” entirely.

- 3) *Abstract line 1 – should be 'have' not 'has'.*

Reply: Corrected in the revised manuscript.

- 4) *Abstract last line: "Hence, a spontaneous dopamine-dependent memory consolidation-based mechanism underlies the long-term behavioral effects of fear extinction." The results suggest this might be the case but do not categorically prove it – I find this phrasing too definitive, I'd soften it a bit 'might underlie'.*

Reply: We agree with this view and have changed the phrasing in the abstract accordingly.

Page 2, Line 17: “Hence, a spontaneous dopamine-dependent memory consolidation-based mechanism may underlie the long-term behavioral effects of fear extinction.”

- 5) *Pg 10, line 9 'Spontaneous vmPFC activity' – do you mean infralimbic?*

Reply: This was indeed incorrect. 'Spontaneous vmPFC activity' was now replaced with:

Page 10, Line 9: 'Spontaneous activity in the infralimbic cortex'.

- 6) *Please can you elaborate on how participants were excluded for 'skin conductance non responding'*

Reply: We have now added greater detail in the description of the exclusion of skin conductance non-responders in the Methods part of the revised manuscript. Please note that the potential exclusion of skin conductance non-responders would always occur *before* the actual start of the experiment. In the present study no potential participant was excluded due to showing no skin conductance response during the screening session.

Page 18: During the screening session we tested whether participants showed normal skin conductance responding. To this aim, we attached two electrodes of the eSense Skin response device (Mindfield® Biosystems Ltd., Berlin, Germany) to the medial phalanges of the first and the third finger. Participants were then asked to take several deep breaths. In addition, we induced a light acoustic startle response by clapping the hands without announcement. Both deep breathing and acoustic startle usually result in a deflection of the skin conductance, not seen in skin conductance non-responding individuals. None of the participants screened for the present study had to be excluded according to these criteria.

- 7) *Similarly, please elaborate on how 'first response onset' is defined for: "SCRs were scored offline as the difference between first response onset in a time window from 900 to 4000 ms after CS onset and the subsequent peak, using a custom-made analysis script."*

Reply: We want to apologize for the lack of detail. We manually score the first local minimum in the skin conductance time course in a window from 900 ms to 4000 ms after stimulus onset as response onset. The, thereupon, following local maximum is scored as "subsequent peak" and the peak-onset difference is assessed as CS evoked skin conductance response. Importantly, the experimenter scoring the data is always blinded in respect to the stimulus type (CS+/CS-) of each SCR and the group belongingness (placebo/L-DOPA) of each participant. We have now included this information in the manuscript (p. 21, l. 18).

Page 21, line 18: Using a custom made analysis script we manually scored the first local minimum in the skin conductance time course in a window from 900 ms to 4000 ms after CS onset as response onset. The, thereupon, following local maximum was scored as response peak and skin conductance responses as peak to onset difference. Importantly, the experimenter scoring the data was blinded to the stimulus type (CS+/CS-) of each SCR and the group belongingness (placebo/L-DOPA) of each participant.

- 8) *"In order to account for physiological noise and harmonize measures across the different experimental phases, we operationalized initial fear acquisition as SCRs to CSs averaged across the last 20% of trials on day 1...". How does this account for physiological noise? Please explain.*

Reply: Single trial SCRs can easily be affected by factors other than the stimulus-evoked response, e.g. concurrent breathing, a stimulus unrelated movement or inattentiveness during an individual trial. To achieve a more reliable measure of the actual stimulus-evoked SCRs, we average SCRs across several trials aiming to reduce the effect of physiological noise. We have now included this motivation in the Methods section of the manuscript (p. 22, l. 15).

Page 22, line 15: As single trial SCRs can easily be affected by factors other than the stimulus-evoked response, e.g. concurrent breathing, a stimulus unrelated movement, we averaged SCRs across several trials to achieve a more reliable measure of the actual stimulus-evoked SCRs.

References

- Baron RM, Kenny DA (1986). The moderator-mediator variable distinction in social psychological research: conceptual, strategic, and statistical considerations. *J Pers Soc Psychol* **51**: 1173–1182.
- Contin M, Martinelli P (2010). Pharmacokinetics of levodopa. *J Neurol* **257**: S253-261.
- Haaker J, Gaburro S, Sah A, Gartmann N, Lonsdorf TB, Meier K, *et al* (2013). Single dose of L-dopa makes extinction memories context-independent and prevents the return of fear. *Proc Natl Acad Sci U S A* **110**: E2428-2436.
- Haaker J, Lonsdorf TB, Kalisch R (2015). Effects of post-extinction I-DOPA administration on the spontaneous recovery and reinstatement of fear in a human fMRI study. *Eur Neuropsychopharmacol* **25**: 1544–1555.
- Hermans EJ, Kanen JW, Tambini A, Fernández G, Davachi L, Phelps EA (2016). Persistence of Amygdala-Hippocampal Connectivity and Multi-Voxel Correlation Structures During Awake Rest After Fear Learning Predicts Long-Term Expression of Fear. *Cereb Cortex N Y N 1991* doi:10.1093/cercor/bhw145.
- Kalisch R, Wiech K, Critchley HD, Dolan RJ (2006). Levels of appraisal: a medial prefrontal role in high-level appraisal of emotional material. *NeuroImage* **30**: 1458–1466.
- Kalisch R, Wiech K, Critchley HD, Seymour B, O'Doherty JP, Oakley DA, *et al* (2005). Anxiety reduction through detachment: subjective, physiological, and neural effects. *J Cogn Neurosci* **17**: 874–883.
- LeWitt PA (2015). Levodopa therapy for Parkinson's disease: Pharmacokinetics and pharmacodynamics. *Mov Disord Off J Mov Disord Soc* **30**: 64–72.
- Lonsdorf TB, Haaker J, Fadai T, Kalisch R (2014). No evidence for enhanced extinction memory consolidation through noradrenergic reuptake inhibition-delayed memory test and reinstatement in human fMRI. *Psychopharmacology (Berl)* **231**: 1949–1962.
- Mumford JA, Davis T, Poldrack RA (2014). The impact of study design on pattern estimation for single-trial multivariate pattern analysis. *NeuroImage* **103**: 130–138.
- Mumford JA, Turner BO, Ashby FG, Poldrack RA (2012). Deconvolving BOLD activation in event-related designs for multivoxel pattern classification analyses. *NeuroImage* **59**: 2636–2643.
- Nyholm D, Lewander T, Gomes-trolin C, Bäckström T, Panagiotidis G, Ehrnebo M, *et al* (2012). Pharmacokinetics of Levodopa/carbidopa Microtablets Versus Levodopa/benserazide and Levodopa/carbidopa in Healthy Volunteers. *Clin Neuropharmacol* **35**: 111–117.
- Raczka KA, Mechias M-L, Gartmann N, Reif A, Deckert J, Pessiglione M, *et al* (2011). Empirical support for an involvement of the mesostriatal dopamine system in human fear extinction. *Transl Psychiatry* **1**: e12.
- Song X-W, Dong Z-Y, Long X-Y, Li S-F, Zuo X-N, Zhu C-Z, *et al* (2011). REST: a toolkit for resting-state functional magnetic resonance imaging data processing. *PloS One* **6**: e25031.
- Staresina BP, Alink A, Kriegeskorte N, Henson RN (2013). Awake reactivation predicts memory in humans. *Proc Natl Acad Sci U S A* **110**: 21159–21164.
- Visser RM, Haan MIC de, Beemsterboer T, Haver P, Kindt M, Scholte HS (2016). Quantifying learning-dependent changes in the brain: Single-trial multivoxel pattern analysis requires slow event-related fMRI. *Psychophysiology* **53**: 1117–1127.

REVIEWERS' COMMENTS:

Reviewer #1 (Remarks to the Author):

The authors convincingly showed that results do not hinge on the exact number of trials included during extinction learning and I have no further concerns.

Reviewer #2 (Remarks to the Author):

The authors have convincingly addressed all my concerns. I am pleased to support publication of this manuscript.

Sincerely,

Marijn Kroes
(I sign all reviews)

Reviewer #3 (Remarks to the Author):

I have read the authors' response to both my comments and those of the other reviewers. The authors have done an excellent and thorough job in their response. It is a pleasure to see both fair but thorough reviews from the other reviewers and a comprehensive response from the authors that goes to lengths to address the issues we have raised (how the process should work!)

I just have a few remaining points

Prior major comments:

1) Satisfactorily addressed – thank you

2) Both Reviewer 2 and I queried the authors interpretation of their results in terms of prediction errors when they did not actually mathematically model prediction errors.

See Reviewer 2, point 6: 'The authors state that the extracted vmPFC data reflects prediction error related activity because "Early in the extinction session, US omission is unexpected to the participants and elicits a prediction error..."

My suggestion was to do the mathematical modeling, reviewer 2's suggestion was to drop the reference to prediction errors. The authors state they opt for the latter suggestion and in response to my request argue that SCR data is too noisy to easily model individual trial data. I feel this is a reasonable response (for future work I note that trial-wise pupil dilation data is somewhat cleaner and easier to use for such model-fitting).

The authors response states: 'we have now changed the phrasing in the main text of the revised manuscript and replaced 'prediction error-related vmPFC activity pattern' with 'CS offset-related vmPFC activity pattern' in the presentation of the results. We only refer to the possibility of an involvement of the prediction error in establishing the respective vmPFC activity pattern in the discussion. There we also acknowledge the limitation of our study to not include modeled prediction error time courses there (p. 11, l. 4). We hope that the reviewer agrees with this procedure. Page 11, line 3: Even though not modeled explicitly in the present study, the unexpected omission of the aversive US early in extinction may elicit a prediction error signal ("outcome better than expected") initiating new learning.

The above is all fine. However, the remaining problem is that on page 11, after the line copied above, the authors have left in a whole page of discussion that still discusses their results based on the assumption that they reflect prediction errors. Further, the first sentence of this section (below) is ungrammatical and it is unclear as to whether it is meant to refer to the authors' current results.

'First evidence from a human fMRI study shows that activity in the mesostriatal dopamine system scales with such a prediction error signal during extinction learning³⁴.

If the authors want to drop the prediction error framing of their results and only mention it in passing as a possibility rather than do the modeling necessary to frame these results in terms of prediction errors that is fine by me, but in that case this is way too much discussion of the now-dropped prediction error framing of the result and needs rewriting.

3) The authors explain the methodological choices both I and reviewer 2 flagged as apparently arbitrary as follows:

'At the outset of a series of three-day studies on the effects of a consolidation-enhancing effect of L-DOPA on fear extinction memory currently ongoing in our laboratory – including the present study – we defined the described analysis strategy as an a-priori analysis strategy for all studies independent of the exact number of trials employed in an individual study during fear conditioning, extinction and test, respectively. The intention was to achieve harmonization and comparability across studies as well as to avoid a-posteriori hypothesis formulation.

I commend the authors on establishing clear a-priori methods that they use across studies. I think this is highly important and worth clearly mentioning in the Methods.

I see the authors now state (in the Methods) that

'We operationalized initial fear acquisition as SCRs to CSs averaged across the last 20% of trials on day 1 (i.e. last 2 trials), fear memory recall as SCRs to CSs averaged across the first 20% of trials on day 2 (i.e. first 3 trials) and fear at the end of extinction as SCRs to CSs averaged across the last 20% of trials (i.e. last 3 trials) on day 2. As in previous studies^{17,19}, the effect of LDOPA on extinction memory retrieval was tested on SCRs to CS+ and CS-, averaged across the whole test phase on day 3 (i.e. all 10 trials).'

It would be great if the authors could add here that this procedure is standardized across studies. This could be briefly referenced in the Results section where the authors could also state that if 5 trials are used (to match the fmri analysis) equivalent results are obtained.

The results below (from the response letter) could then be put into the Supplements.

Day 2: test of stimulus and group effect at the beginning of extinction (with first 5 trials: stimulus:

$F_{1,34}=49.65$, $P<.001$, partial $\eta^2=.59$; stimulus x group: $P=.55$; $n=36$).

Day 2: test of stimulus and group effect at the end of extinction (with first 5 trials: stimulus:

$F_{1,34}=23.65$, $P<.001$, partial $\eta^2=.41$; stimulus x group: $P=.78$; $n=36$).

I hope this might be satisfactory and not too disruptive to the flow. I am open to the authors' thoughts on this. Given some studies have supposedly 'a-prior' method choices that seem very post-hoc, I think it would be nice to clearly demonstrate that the choices here were indeed a-priori and cross-study and that the results held regardless.

4) Old comment 'Figure 2c and Fig S2, S3 – please do not use uncorrected thresholds to show small volume activations – please use the corrected thresholds as in the main text. Otherwise the activations displayed are misleading.'

I thank the author for addressing this for Figure 2c. I must ask that they also use corrected thresholds in their figures S2, S3, S6, and S7.

I appreciate the argument that it seems odd to use SVC thresholds to also threshold outside of ROIs but that the authors might wish to show that activations do not extend beyond the ROI. However, that does not seem to be a big issue here as the activations are very focal and I think it is a far bigger problem that use of an uncorrected threshold will end up with false positives being displayed which may mislead readers as to the extent of a given activation. If the authors wish they could simply state they have used an equivalent threshold outside the ROI for illustration purposes (I believe this is the lesser of two evils) or they can mask outside the ROI. Using a

different threshold for analysis of ROI data and display of the data to complement the analysis is I believe likely to be the most misleading option.

Prior minor points: all satisfactorily handled.

New minor points:

1) The authors use the term 'trend-wise significant effect' (e.g. in the legend for Fig S6) for effects that are between $p < .05$ and $p < .1$. These should just be referred to as trends or near-significant trends. It is confusing to refer to them as significant when a threshold of $p < .05$ is being adopted (as is typically done).

2) Fig 1 legend. The authors have added that 'The groups differed significantly on mean SCRs across the test phase on day 3' and 'Note, that there was only a significant difference between groups on mean SCRs across the test phase' in order to indicate that there is no interaction with time / change across trials.

However this leaves it unclear as to whether it is the mean SCR across both CS+ and CS- averaged together they are referring to, or whether the groups differ in the average SCR for the CS+ alone (and not to the CS-) or if they are referencing the average response to the CS+ versus CS- (given the key significant interaction of drug x stimulus type indicated in the main text). It would be helpful if this could be clarified.

Typos/proof edits:

Major

Fig S5 legend. The new text appears to have a number of cut and paste errors (astrixed) (d) Critically, the effects are also not observed for reactivations (45 min post-extinction) of the first 5 CS+ onset-related vmPFC activity pattern (linear regression: $\beta_{45\text{min}} = -.03$, $SE = .03$, $T_{33} = -.79$, $P = .43$; $n = 35$). There was also no relation between spontaneous *CS- offset-related* vmPFC reactivations at 10 or 90 minutes after extinction learning and CR at test (data not shown, $P_s > .07$) [I believe this should also be first 5 CS+ onset-related]... (f) Finally, the effects are also not observed for reactivations (45 min post-extinction) of the *middle* 5 CS+ onset-related vmPFC activity pattern (linear regression: $\beta_{45\text{min}} = -.03$, $SE = .04$, $T_{33} = -.66$, $P = .51$; $n = 35$). There was no relation between spontaneous *middle* 5 CS+ onset-related vmPFC reactivations at 10 or 90 minutes after extinction learning and CR at test (data not shown, $P_s > .12$) [I believe this should be last].

Minor

a) Intro: highlighted text 'These results provide first evidence' . Also Discussion, page 12 'Our results extend these findings by providing first evidence' and pg 13 'the present study further provides first evidence'

This should be 'the first evidence' or 'initial evidence' to be grammatical

b) Pg 7 'comparable relations' (para 2), 'negative relation' (para 3), 'the relation' (twice in para 4) I think this should be 'relationship(s) not relation(s)

c) Pg 7 'Lastly, the results are not dependent on clearing or not-clearing the resting-state BOLD activity time courses from nuisance signals (i.e. cerebrospinal fluid, white matter, head motion) before identifying potential vmPFC reactivations (Supplementary Figure 7)

This would more typically be phrased as cleaning, or not cleaning, nuisance signals (i.e. cerebrospinal fluid, white matter, head motion) from the resting-state BOLD activity time courses before identifying potential vmPFC reactivations (Supplementary Figure 7)

d) Pg 8 10.000 should be written as 10,000

e) Pg 9. Confirming previous work in rodents^{17,18} and humans¹⁷ we observed that a post-extinction L-DOPA administration enhanced extinction memory retrieval relative to placebo administration

I think it might be more appropriate to say 'in line with' rather than 'confirming'

f) Pg 9. When the authors say 'In addition to these previous studies, we can now show that the effect of LDOPA on extinction memory retrieval'

Are they saying this is the aspect that goes beyond the findings of the prior studies, i.e. the novel additional component to the findings here? If so it might be clearer to rephrase this 'Our findings extend beyond those reported to date by showing that',

g) Methods pg 24.

Even though CS onset- and CS offset-related regressors succeeded each other by 4.5 seconds only, correlations between the HRF-convolved regressors were low (Supplementary Figure 9).

In the above sentence convolved is mis-spelt. I also think it would make more sense to put this sentence after the following one as it references HRF-convolution

Following sentence: 'All regressors were modeled as delta-functions and convolved with the canonical hemodynamic response function (HRF).'

Reviewer #1 (Remarks to the Author):

The authors convincingly showed that results do not hinge on the exact number of trials included during extinction learning and I have no further concerns.

Reviewer #2 (Remarks to the Author):

*The authors have convincingly addressed all my concerns. I am pleased to support publication of this manuscript.
Sincerely,
Marijn Kroes*

(I sign all reviews)

Reviewer #3 (Remarks to the Author):

I have read the authors' response to both my comments and those of the other reviewers. The authors have done an excellent and thorough job in their response. It is a pleasure to see both fair but thorough reviews from the other reviewers and a comprehensive response from the authors that goes to lengths to address the issues we have raised (how the process should work!)

I just have a few remaining points

Prior major comments:

1) Satisfactorily addressed – thank you

2) Both Reviewer 2 and I queried the authors interpretation of their results in terms of prediction errors when they did not actually mathematically model prediction errors.

See Reviewer 2, point 6: 'The authors state that the extracted vmPFC data reflects prediction error related activity because "Early in the extinction session, US omission is unexpected to the participants and elicits a prediction error..."

My suggestion was to do the mathematical modeling, reviewer 2's suggestion was to drop the reference to prediction errors. The authors state they opt for the latter suggestion and in response to my request argue that SCR data is too noisy to easily model individual trial data. I feel this is a reasonable response (for future work I note that trial-wise pupil dilation data is somewhat cleaner and easier to use for such model-fitting). The authors response states: 'we have now changed the phrasing in the main text of the revised manuscript and replaced 'prediction error-related vmPFC activity pattern' with 'CS offset-related vmPFC activity pattern' in the presentation of the results. We only refer to the possibility of an involvement of the prediction error in establishing the respective vmPFC activity pattern in the discussion. There we also acknowledge the limitation of our study to not include modeled prediction error time courses there (p. 11, l. 4). We hope that the reviewer agrees with this procedure. Page 11, line 3: Even though not modeled explicitly in the present study, the unexpected omission of the aversive US early in extinction may elicit a prediction error signal ("outcome better than expected") initiating new learning.

The above is all fine. However, the remaining problem is that on page 11, after the line copied above, the authors have left in a whole page of discussion that still discusses their results based on the assumption that they reflect prediction errors. Further, the first sentence of this section (below) is ungrammatical and it is unclear as to whether it is meant to refer to the authors' current results.

'First evidence from a human fMRI study shows that activity in the mesostriatal dopamine system scales with such a prediction error signal during extinction learning³⁴.

If the authors want to drop the prediction error framing of their results and only mention it in passing as a possibility rather than do the modeling necessary to frame these results in terms of prediction errors that is fine by me, but in that case this is way too much discussion of the now-dropped prediction error framing of the result and needs rewriting.

Reply: We thank the reviewer for raising this concern. We fully appreciate the reviewer's argument that the length of the respective paragraph does not accurately reflect the lack of evidence for an involvement of the

prediction error signal in the formation of the vmPFC activity pattern. We have therefore shortened the respective paragraph and now dedicate more space to this actual limitation of our study.

Page 11, line 5: Our results show that specifically reactivations of the vmPFC activity pattern elicited at the time of the unexpected omission of the US at CS+ offset early in extinction predict extinction memory retrieval. Why may this specific vmPFC pattern be important for extinction memory retrieval? The unexpected omission of the aversive US early in extinction may elicit a surprise, relief or prediction error (“outcome better than expected”) signal. It is conceivable that upon such a signal the vmPFC acquires and updates a representation of the expected value of the new “CS-noUS” association, in line with the role of the vmPFC in expected value representation³⁴. A limitation of our study is that we did not model the prediction error signal explicitly (due to the limited reliability of model parameters estimated based on noisy trial-by-trial SCR). Thus, future work is necessary to elucidate the contribution of prediction error signals to the formation of the CS+ offset-related vmPFC activity pattern, whose post-learning reactivation supports extinction memory consolidation.

3) *The authors explain the methodological choices both I and reviewer 2 flagged as apparently arbitrary as follows:*

'At the outset of a series of three-day studies on the effects of a consolidation-enhancing effect of L-DOPA on fear extinction memory currently ongoing in our laboratory – including the present study – we defined the described analysis strategy as an a-priori analysis strategy for all studies independent of the exact number of trials employed in an individual study during fear conditioning, extinction and test, respectively. The intention was to achieve harmonization and comparability across studies as well as to avoid a-posteriori hypothesis formulation.

I commend the authors on establishing clear a-priori methods that they use across studies. I think this is highly important and worth clearly mentioning in the Methods.

I see the authors now state (in the Methods) that

'We operationalized initial fear acquisition as SCRs to CSs averaged across the last 20% of trials on day 1 (i.e. last 2 trials), fear memory recall as SCRs to CSs averaged across the first 20% of trials on day 2 (i.e. first 3 trials) and fear at the end of extinction as SCRs to CSs averaged across the last 20% of trials (i.e. last 3 trials) on day 2. As in previous studies^{17,19}, the effect of LDOPA on extinction memory retrieval was tested on SCRs to CS+ and CS-, averaged across the whole test phase on day 3 (i.e. all 10 trials).'

It would be great if the authors could add here that this procedure is standardized across studies.

This could be briefly referenced in the Results section where the authors could also state that if 5 trials are used (to match the fmri analysis) equivalent results are obtained.

The results below (from the response letter) could then be put into the Supplements.

Day 2: test of stimulus and group effect at the beginning of extinction (with first 5 trials: stimulus: $F_{1,34}=49.65$, $P<.001$, partial $\eta^2=.59$; stimulus x group: $P=.55$; $n=36$).

Day 2: test of stimulus and group effect at the end of extinction (with first 5 trials: stimulus: $F_{1,34}=23.65$, $P<.001$, partial $\eta^2=.41$; stimulus x group: $P=.78$; $n=36$).

I hope this might be satisfactory and not too disruptive to the flow. I am open to the authors' thoughts on this. Given some studies have supposedly 'a-prior' method choices that seem very post-hoc, I think it would be nice to clearly demonstrate that the choices here were indeed a-priori and cross-study and that the results held regardless.

Reply: We fully agree with the reviewer that a post-hoc justification of such choices should be prevented. We have now included information about the standardization of this procedure across a series of experiments from our laboratory in the respective paragraph of the Methods section. We also included the mentioned results in the Supplementary Information. In order to not interrupt the concise description of the SCR Results in the main text we have included the reference to these additional results in the above mentioned paragraph of the Methods section and hope that the reviewer agrees with this decision.

Page 22, line 19: As single-trial SCRs can easily be affected by factors other than the stimulus-evoked response (e.g. concurrent breathing, a stimulus unrelated movement), we averaged SCRs across several trials to achieve a more reliable measure of the actual stimulus-evoked SCRs. In order to harmonize measures across a series of experiments on the effect of L-DOPA on extinction memory consolidation with varying trial numbers (Gerlicher et al., in preparation), we standardized the operationalization of fear acquisition, fear memory recall, fear at the end of extinction and extinction memory retrieval a-priori across studies. Namely, we operationalized initial fear acquisition as SCRs to CSs averaged across the last 20% of trials on day 1 (here: last 2 trials), fear memory recall as SCRs to CSs averaged across the first 20% of trials on day 2 (here: first 3 trials) and fear at the end of extinction as SCRs to CSs averaged across the last 20% of trials (here: last 3 trials) on day 2. Note, that instead employing the first 5 and last 5 trials during extinction, as in the fMRI analysis, does not change the results (Supplementary Table 2).

Supplementary Table 2. The results of the analysis of the SCR during extinction learning on day 2 do not hinge on the exact choice of trials employed for assessing CR at the beginning or end of extinction. That is, when operationalizing CR at the beginning of extinction as CS+/CS- evoked SCR averaged across the first 5 trials (as in the fMRI analysis) instead of the first 20% of trials (i.e. 3 trials) as in all SCR analyses (see Methods) we still find a significant effect of stimulus that does not significantly differ between groups. Similarly, operationalizing CR at the end of extinction as CS+/CS- evoked SCR averaged across the last 5 trials (as in the fMRI analysis), we also find a significant effect of stimulus that does not differ between groups.

	Main effect stimulus	Main effect group	Interaction effect stimulus x group
CR at beginning of ext. (mean CS+/CS- across first 5 trials)	$F_{1,34} = 49.65$ $P < .001$ partial $\eta^2 = .59$	$F_{1,34} = .36$ $P = .55$	$F_{1,34} = .37$ $P = .55$
CR at the end of ext. (mean CS+/CS- across last 5 trials)	$F_{1,34} = 23.65$ $P < .001$ partial $\eta^2 = .41$	$F_{1,34} = .12$ $P = .73$	$F_{1,34} = .08$ $P = .78$

4) *Old comment 'Figure 2c and Fig S2, S3 – please do not use uncorrected thresholds to show small volume activations – please use the corrected thresholds as in the main text. Otherwise the activations displayed are misleading.'*

I thank the author for addressing this for Figure 2c. I must ask that they also use corrected thresholds in their figures S2, S3, S6, and S7.

I appreciate the argument that it seems odd to use SVC thresholds to also threshold outside of ROIs but that the authors might wish to show that activations do not extend beyond the ROI. However, that does not seem to be a big issue here as the activations are very focal and I think it is a far bigger problem that use of an uncorrected threshold will end up with false positives being displayed which may mislead readers as to the extent of a given activation. If the authors wish they could simply state they have used an equivalent threshold outside the ROI for illustration purposes (I believe this is the lesser of two evils) or they can mask outside the ROI. Using a different threshold for analysis of ROI data and display of the data to complement the analysis is I believe likely to be the most misleading option.

Reply: We followed the reviewer's suggestion to employ SVC corrected statistical thresholds for the graphical illustration of the fMRI results and have updated Figure 2c in the revised versions of the Manuscript and Figure S2, Figure S3, Figure S6 and Figure S7 in the Supplementary Information.

Prior minor points: all satisfactorily handled.

New minor points:

1) The authors use the term 'trend-wise significant effect' (e.g. in the legend for Fig S6) for effects that are between $p < .05$ and $p < .1$. These should just be referred to as trends or near-significant trends. It is confusing to refer to them as significant when a threshold of $p < .05$ is being adopted (as is typically done).

Reply: We have now changed the respective phrasing both in the Manuscript and the Supplementary Information.

Page 8, line 11: Indeed, post-extinction L-DOPA administration significantly increased the number of potential CS+ offset-related vmPFC pattern reactivations (time x group: $F_{3,114}=3.51$, $P=.02$, partial $\eta^2=.09$; $n=40$), particularly 45 minutes after extinction and still a near-significant trend at 90 minutes after extinction (post-hoc two-sample t test; pre: $P=.39$; post: 10 min: $P=.09$; 45 min: $T_{38}=-3.05$, $P=.004$, Cohen's $d=.97$; 90 min: $T_{38}=-1.95$, $P=.06$, Cohen's $d=.62$; $n=40$; Fig. 2e).

Legend Figure S3 (SI, p. 5): In addition, the exact threshold definition did not affect the mediation of the effect of L-DOPA on CR at test by the number of potential spontaneous CS+ offset-related vmPFC pattern reactivations using a more conservative threshold (with $Z > 2.25$; mediation analysis: $\beta = -.06$, 95% CI: $-.12$ -. $.01$, $P=.04$; $n=35$), even though this effect was only a near-significant trend when using the more liberal threshold (with $Z > 1.65$; mediation analysis: $\beta = -.05$, 95% CI: $-.10$ -. $.00$, $P=.06$; $n=35$). Data are presented as mean \pm s.e.m.

Legend Figure S6 (SI, p. 10): **g)** the first 4 CS+ offsets ($x,y,z=4,34,-20$; $Z=3.61$, $P=.02$, SVC, FWE; $n=40$), **h)** the first 6 CS+ offsets ($x,y,z=6,46,-14$; $Z=5.34$, $P<.001$, SVC, FWE; $n=40$), and **i)** the first 7 CS+ offsets ($x,y,z=8,46,-14$; $Z=3.90$, $P=.009$, SVC, FWE; $n=40$), but only a near-significant trend for **j)** the first 8 CS+ offsets ($x,y,z=6,44,-14$; $Z=3.27$, $P=.061$, SVC, FWE; $n=40$). Display threshold $P<.001$, uncorr., no masking applied. **k)** There was no significant effect of L-DOPA on number of vmPFC activity pattern reactivations computed based on the first 3 CS+ offsets (repeated-measures ANOVA, time x group: $F_{3,114}=4.48$, $P=.07$; $n=40$). **l)** There was a near-significant trend towards a greater number of vmPFC activity pattern reactivations computed based on the first 4 CS+ offsets after L-DOPA intake (repeated-measures ANOVA, time x group: $F_{3,114}=2.30$, $P=.08$; $n=40$) specifically 45 min after extinction (post-hoc two-sample t tests: pre: $P=.12$; post: 10 min: $P=.34$; 45 min: $T_{38}=-1.95$, $P=.06$; 90 min: $T_{38}=-1.86$, $P=.07$; $n=40$). **m)** There was a near-significant main effect of drug independent of time on the first 6 CS+ offsets (repeated-measures ANOVA, group: $F_{1,38}=3.66$, $P=.06$; $n=40$) due to significantly more vmPFC reactivations 45 min after extinction in L-DOPA treated participants (post-hoc two-sample t tests: pre: $P=.42$; post: 10 min: $P=.20$; 45 min: $T_{38}=-2.09$, $P=.04$, Cohen's $d=.67$; 90 min: $T_{38}=-1.46$, $P=.15$; $n=40$). **n)** The number of reactivations of the first 7 CS+ offset vmPFC activity pattern was significantly greater in the L-DOPA group (repeated-measures ANOVA, group: $F_{1,38}=4.39$, $P=.04$, partial $\eta^2=.07$; $n=40$), due to an effect of L-DOPA on number of vmPFC reactivations 45 min after extinction (post-hoc two-sample t tests: pre: $P=.78$; post: 10 min: $P=.50$; 45 min: $T_{38}=-2.32$, $P=.03$, Cohen's $d=.71$; 90 min: $T_{38}=-1.90$, $P=.06$; $n=40$). **o)** Lastly, on the first 8 CS+ offsets there was a near-significant time by group interaction (repeated-measures ANOVA, time x group: $F_{3,114}=3.10$, $P=.06$, Greenhouse-Geisser corrected) with L-DOPA treated participants showing significantly more vmPFC reactivations specifically 90 min after extinction (post-hoc two-sample t tests: pre: $P=.39$; post: 10 min: $P=.73$; 45 min: $P=.12$; 90 min: $T_{38}=-3.11$, $P=.004$, Cohen's $d=.97$; $n=40$).

Legend Figure S7 (SI, p. 11): **(c)** There was a near-significant trend towards an effect of drug group on number of CS+ offset-related vmPFC reactivations (repeated-measures ANOVA, group: $F_{1,38}=3.99$, $P=.05$, partial $\eta^2=.10$; $n=40$), due to significantly greater number of reactivations in L-DOPA compared to placebo treated participants 45 min after extinction (two-sample T -test: post-hoc t tests: pre: $P=.66$; post: 10 min: $P=.17$; 45 min: $T_{36}=-2.42$, $P=.02$, Cohen's $d=.77$; 90 min: $P=.40$; $n=40$).

Legend Figure S8 (SI, p. 12): Note, that analysis of simple slopes in each group indicated, though, that the relationship between CRs at the end of extinction and CRs at test showed a near-significant trend towards a positive relationship after L-DOPA ($\beta_{L-DOPA}=.29$, $SE=.15$, $T_{29}=1.93$, $P=.08$), but not after placebo administration ($\beta_{placebo}=.15$, $SE=.13$, $T_{29}=1.17$, $P=.25$).

2) Fig 1 legend. The authors have added that 'The groups differed significantly on mean SCRs across the test phase on day 3' and 'Note, that there was only a significant difference between groups on mean SCRs across the test phase' in order to indicate that there is no interaction with time / change across trials. However this leaves it unclear as to whether it is the mean SCR across both CS+ and CS- averaged together they are referring to, or whether the groups differ in the average SCR for the CS+ alone (and not to the CS-) or if

they are referencing the average response to the CS+ versus CS- (given the key significant interaction of drug x stimulus type indicated in the main text). It would be helpful if this could be clarified.

Reply: We agree with the reviewer that the description of the results is unclear. We have now revised it, as follows:

Page 14, line 14: Note, that the group difference stemmed from significantly smaller CS+ evoked SCRs averaged across the whole test phase, but the speed of re-extinction did not differ significantly between drug groups (control analysis with stimulus (CS+, CS-) and trial (1-10) as within-, and drug group (placebo, L-DOPA) as between-subject factor: stimulus x group: $F_{1,297}=6.57$, $P=.02$, partial $\eta^2=.17$; stimulus x trial x group: $F_{1,297}=1.32$, $P=.23$; $n=35$).

Typos/proof edits:

Major

*Fig S5 legend. The new text appears to have a number of cut and paste errors (astrixed) (d) Critically, the effects are also not observed for reactivations (45 min post-extinction) of the first 5 CS+ onset-related vmPFC activity pattern (linear regression: $\beta_{45min}=-.03$, $SE=.03$, $T_{33}=-.79$, $P=.43$; $n=35$). There was also no relation between spontaneous *CS- offset-related* vmPFC reactivations at 10 or 90 minutes after extinction learning and CR at test (data not shown, $P_s>.07$) [I believe this should also be first 5 CS+ onset-related]... (f) Finally, the effects are also not observed for reactivations (45 min post-extinction) of the *middle* 5 CS+ onset-related vmPFC activity pattern (linear regression: $\beta_{45min}=-.03$, $SE=.04$, $T_{33}=-.66$, $P=.51$; $n=35$). There was no relation between spontaneous *middle* 5 CS+ onset-related vmPFC reactivations at 10 or 90 minutes after extinction learning and CR at test (data not shown, $P_s>.12$) [I believe this should be last].*

Reply: Thank you very much for pointing out these mistakes. The legend was indeed incorrect, we have now included a corrected version in the revised Supplementary Information.

Legend Figure S5 (SI, p. 5): (d) Critically, the effects are also not observed for reactivations (45 min post-extinction) of the first 5 CS+ onset-related vmPFC activity pattern (linear regression: $\beta_{45min}=-.03$, $SE=.03$, $T_{33}=-.79$, $P=.43$; $n=35$). There was also no relationship between spontaneous reactivations of the first 5 CS+ onset-related vmPFC activity pattern at 10 or 90 minutes after extinction learning and CR at test (data not shown, $P_s>.07$). [...] **(f)** Finally, the effects are also not observed for reactivations (45 min post-extinction) of the last 5 CS+ onset-related vmPFC activity pattern (linear regression: $\beta_{45min}=-.03$, $SE=.04$, $T_{33}=-.66$, $P=.51$; $n=35$). There was no relationship between spontaneous last 5 CS+ onset-related vmPFC reactivations at 10 or 90 minutes after extinction learning and CR at test (data not shown, $P_s>.12$).

Minor

a) Intro: highlighted text 'These results provide first evidence' . Also Discussion, page 12 'Our results extend these findings by providing first evidence' and pg 13 'the present study further provides first evidence' This should be 'the first evidence' or 'initial evidence' to be grammatical

Reply: We have changed the sentences accordingly.

b) Pg 7 'comparable relations' (para 2), 'negative relation' (para 3), 'the relation' (twice in para 4) I think this should be 'relationship(s) not relation(s)

Reply: Adopted in the revised version of the Manuscript and the Supplementary Information.

c) Pg 7 'Lastly, the results are not dependent on clearing or not-clearing the resting-state BOLD activity time courses from nuisance signals (i.e. cerebrospinal fluid, white matter, head motion) before identifying potential vmPFC reactivations (Supplementary Figure 7)

This would more typically be phrased as cleaning, or not cleaning, nuisance signals (i.e. cerebrospinal fluid, white matter, head motion) from the resting-state BOLD activity time courses before identifying potential vmPFC reactivations (Supplementary Figure 7)

Reply: Replaced in the revised version of the Manuscript.

Page 7, line 33: Lastly, the results are not dependent on cleaning or not-cleaning the resting-state BOLD activity time courses from nuisance signals (i.e. cerebrospinal fluid, white matter, head motion) before identifying potential vmPFC reactivations (Supplementary Figure 7).

d) Pg 8 10.000 should be written as 10,000

Reply: Corrected in the revised version of the Manuscript.

e) Pg 9. *Confirming previous work in rodents^{17,18} and humans¹⁷ we observed that a post-extinction L-DOPA administration enhanced extinction memory retrieval relative to placebo administration*

I think it might be more appropriate to say 'in line with' rather than 'confirming'

Reply: Adopted in the revised version of the Manuscript.

f) Pg 9. *When the authors say 'In addition to these previous studies, we can now show that the effect of LDOPA on extinction memory retrieval'*

Are they saying this is the aspect that goes beyond the findings of the prior studies, i.e. the novel additional component to the findings here? If so it might be clearer to rephrase this 'Our findings extend beyond those reported to date by showing that'

Reply: We agree with the reviewer that the suggested expression is more precise and have changed the sentence accordingly.

Page 9, line 32: Our findings extend beyond those reported to date by showing that the effect of L-DOPA on extinction memory retrieval was mediated by the L-DOPA-induced increase in spontaneous post-extinction vmPFC reactivations.

g) *Methods pg 24.*

Even though CS onset- and CS offset-related regressors succeeded each other by 4.5 seconds only, correlations between the HRF-convolved regressors were low (Supplementary Figure 9).

In the above sentence convolved is mis-spelt. I also think it would make more sense to put this sentence after the following one as it references HRF-convolution

Following sentence: 'All regressors were modeled as delta-functions and convolved with the canonical hemodynamic response function (HRF).'

Reply: We have corrected the spelling mistake and changed the order of the sentences.

Page 24, line 20: All regressors were modeled as delta-functions and convolved with the canonical hemodynamic response function (HRF). Even though CS onset- and CS offset-related regressors succeeded each other by 4.5 seconds only, correlations between the HRF-convolved regressors were low (Supplementary Figure 9).